# PASER: Post-Training Data Selection for Efficient Pruned Large Language Model Recovery

## Abstract

Model pruning is an effective approach for compressing Large Language Models (LLMs) and improving inference efficiency. However, this process often leads to significant degradation of model capabilities. While post-training techniques such as instruction tuning are commonly employed to recover model performance, existing methods often overlook the uneven deterioration of model capabilities and incur high computational costs due to extensive recovery training. Moreover, some instruction data irrelevant to model capability recovery may introduce negative effects. To address these challenges, we propose the **P**ost-training d**A**ta **S**election method for **E**fficient pruned large language model **R**ecovery (**PASER**). PASER aims to identify instructions where model capabilities are most severely compromised within a certain recovery data budget. Our approach first applies manifold learning and spectral clustering to group recovery data in the semantic space, revealing capability-specific instruction sets. We then adaptively allocate the data budget to different clusters based on the degrees of model capability degradation. In each cluster, we prioritize data samples where model performance has declined dramatically. To mitigate potential negative transfer, we also detect and filter out conflicting or irrelevant recovery data. Extensive experiments demonstrate that PASER significantly outperforms conventional baselines, effectively recovering the general capabilities of pruned LLMs while utilizing merely 4%-20% of the original post-training data and substantially reducing training computational overhead.

## 1 Introduction

Model pruning, which aims at reducing the model size while maintaining or enhancing model capabilities, has been a promising approach towards large language model (LLM) compression and efficient inference. Mainstream LLM pruning schemes include unstructured (Frantar & Alistarh, 2023), semi-structured (Sun et al., 2024), and structured pruning (Ma et al., 2023). In practice, the capability degradation of the pruned model compared with the original LLM is almost unavoidable, especially under high pruning ratios. This degradation phenomenon is more severe for the structured pruning scheme, which has been widely adopted in industrial LLM compression thanks to its hardware-friendly property. Therefore, first *pruning*, then *post-training* has been one of the standard pipelines, where the recovery post-training is incorporated to restore the critical capabilities and adapt to the reduced model capacity (Ma et al., 2023; Zhao et al.).

While various types of post-training data, including pre-training corpora and extensive fine-tuning datasets, have been explored (Xia et al., 2024; Sun et al., 2024), instruction tuning data has demonstrated unique advantages for efficient capability recovery (Ma et al., 2023; Zhao et al.; Zhang et al., 2024; Chen et al., 2023). Compared to pre-training that requires massive computational resources, instruction tuning enables effective recovery with much smaller data scale through explicit supervision. Besides, instruction tuning helps maintain the model's general-purpose abilities through diverse task coverage while avoiding overly specialized recovery.

Conventional schemes (Ma et al., 2023) usually employ the full version of publicly available instruction tuning datasets like Alpaca (Taori et al., 2023) to conduct the recovery post-training. However, this can bring significant computation overhead and even unsatisfactory recovery performance, as shown in Figure 1. An intuitive solution is to take part of the original data for training, thus consuming less data and correspondingly reducing the computation resource demand. Nevertheless,

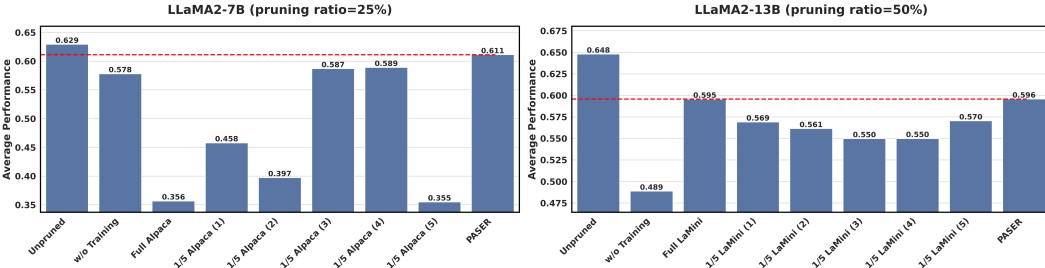

Figure 1: Average performance on seven common LLM reasoning evaluation tasks after recovery post-training with different data. The numbers in brackets represent the group index of the data subset in the full dataset. *Unpruned* indicates the original model and *w/o Training* indicates the pruned model (using LLM-Pruner (Ma et al., 2023)) without the recovery post-training.

directly utilizing the uniformly split data subset (e.g., first 20% of the data), can lead to sub-optimal performance, or even performance degradation. Moreover, the recovered performance considerably varies for models trained with different subsets. Therefore, selecting the most valuable instruction-tuning data that can contribute to recovery performance and reduce training costs becomes crucial. Though previous works have noticed the significance of selecting high-quality data to conduct the general instruction tuning (Wang et al., 2024), few of them are specifically designed for the recovery post-training. Note that general high quality does not necessarily mean useful for recovery.

Considering above limitations, the ideal recovery training data selection approach should exhibit following properties: (1) **Targeted and Balanced Capability Recovery**: The ideal selection method should address the disproportionate impact of pruning on various model capabilities by enabling targeted recovery of critically affected ones, while maintaining a balanced approach overall. It should curate a diverse instruction set that prioritizes severely impacted capabilities without neglecting holistic model functionality. (2) **Recovery Training Efficiency**: Limited computing resources pose serious efficiency challenges to the LLM recovery post-training. An ideal method should be able to select instructions that are both most beneficial for recovery and low in computational cost, thereby accelerating the recovery training process and optimizing resource utilization. (3) **Mitigation of Negative Transfer**: Recognizing that not all instruction data is beneficial for model recovery, an optimal approach should not only identify the most advantageous instructions, but also filter out potentially harmful or irrelevant ones. This significantly reduces the risk of negative transfer during the recovery training, ensuring that the selected data contributes positively to model performance.

To achieve such goals, we propose the **P**ost-training d**A**ta **S**election method for **E**fficient pruned large language model **R**ecovery (**PASER**). First, we perform semantic-structural recovery instruction clustering to obtain different data groups corresponding to different LLM capabilities. Second, we select the recovery instructions in a capability degradation-aware manner, where the overall data budget is allocated to different clusters based on their corresponding capability degradation degrees. In particular, the computation cost of each sample is also taken into consideration during the inner-capability sample selection. Finally, we construct the concept consistency graph to maintain the semantic consistency across the selected instructions, thus preventing introducing conflicting or irrelevant samples. We take the LLaMA 2/3 and Baichuan2 as the target LLMs and perform the experiments under different LLM pruning schemes and different-sized instruction tuning datasets. The comparison with random and conventional instruction tuning data selection baselines demonstrates that PASER can more effectively enhance the recovered LLM performance on language modeling and various reasoning tasks. Meanwhile, the recovery training overhead can also be reduced significantly.

## 2 RELATED WORKS

**Large Language Model Pruning** can be generally divided into three categories: unstructured pruning, semi-structured pruning, and structured pruning. Unstructured pruning removes individual weights without structural constraints, with representative works including SparseGPT (Frantar & Alistarh, 2023), Wanda (Sun et al., 2024), BESA (Xu et al., 2024b), and OWL (Yin et al., 2024). This technique allows for maximum flexibility in weight selection and can achieve high compression rates while maintaining model performance. However, the resulting irregular sparsity patterns limits

the practical acceleration. Semi-structured pruning (Guo et al., 2024; Malla et al., 2024; Frantar & Alistarh, 2023; Sun et al., 2024) targets specific patterns like N:M sparsity, balancing flexibility and hardware efficiency. Structured pruning approaches like LLM-Pruner (Ma et al., 2023) and SliceGPT (Ashkboos et al., 2024) remove entire structural components, offering better hardware compatibility and attracting industry attention (Ko et al., 2023; An et al., 2024; Song et al., 2024; Xia et al., 2024). However, structured pruning faces more severe performance degradation, highlighting the importance of recovery post-training.

**Instruction Tuning** has emerged as a crucial technique for enhancing LLMs (Wei et al.; Wang et al., 2023), improving their adaptability to novel tasks (Sanh et al.; Liang et al.; Zhou et al., 2024). Recent works have explored instruction tuning as a post-compression recovery mechanism (Zhao et al.; Ma et al., 2023). While promising, this combination faces challenges from reduced model capacity and computational costs. Most current approaches use general instruction datasets without considering compressed model's characteristics or disproportionately affected capabilities. Our work addresses these gaps by proposing a novel framework for post-training data selection in pruned LLM recovery.

## 3 PRELIMINARIES

Let $M_o$ denote the original large language model (LLM) and $M_p$ the pruned version of this model. We define the instruction tuning dataset as $D = \{(x_i, y_i)\}_{i=1}^N$, where $x_i$ represents an instruction and $y_i$ its corresponding output. Our goal is to select a subset $S \subset D$ to efficiently recover the performance of $M_p$. We formulate the problem as an optimization task:

$$S^* = \underset{S \subset D, |S| \leq B}{\arg\min} \mathbb{E}_{(x,y) \sim \mathcal{T}}[\mathcal{L}(M_r(S), x, y)],$$

$$\text{subject to} \quad M_r(S) = \text{RecoveryTrain}(M_p, S) \tag{1}$$

where $M_r(S)$ is the recovered model after training on subset $S$, $\mathcal{T}$ is the distribution of downstream evaluation tasks, $\mathcal{L}$ is a loss function. $B(B < N)$ is the recovery data budget, i.e., maximum number of samples allowed in the selected subset.

## 4 METHODOLOGY

In this section, we introduce the three main components of the PASER framework: semantic-structural recovery instruction clustering, capability degradation-aware adaptive recovery instruction selection, and negative transfer mitigation. Furthermore, we provide the time complexity analysis for PASER in the selection process.

### 4.1 SEMANTIC-STRUCTURAL RECOVERY INSTRUCTION CLUSTERING

During the LLM pruning process, different model capabilities may be affected unevenly by pruning. To ensure balanced recovery, we need to identify and group data points that target similar capabilities. To achieve this, we hypothesize that distinct geometric topological structures of recovery instruction data in the high-dimensional semantic space may correspond to different aspects of LLM capabilities. This hypothesis is based on the intuition that instructions requiring similar capabilities are likely to cluster together in the semantic space, forming identifiable topological structures. In detail, we employ a two-step approach on the embedding space of instructions. First, we apply a diffusion kernel to SentenceBERT (Reimers & Gurevych, 2019) embeddings for manifold learning to uncover the intrinsic geometric structure in the semantic space and reduce dimensionality while preserving non-linear relationships:

$$e(x_i) = \text{DiffusionKernel}(\text{SentenceBERT}(x_i)). \tag{2}$$

Here, $e(x_i)$ is the obtained low-dimensional manifold representation of instruction $x_i$. Then, non-negative matrix factorization(NMF)-based spectral clustering (Ding et al., 2005) is conducted to identify natural groupings of instructions that potentially correspond to different LLM capabilities, leveraging the low-dimensional manifold representation:

$$C = \{c_1, \ldots, c_K\} = \text{NMFSpectralClustering}(\{e(x_i) | (x_i, y_i) \in D\}). \tag{3}$$

As for the details in the first step, we first obtain the SentenceBERT embedding of each instruction. Then, we construct an adjacency matrix $A$ based on the pairwise distances of these embeddings: $A_{ij} = \exp(-\frac{\|\text{SentenceBERT}(x_i) - \text{SentenceBERT}(x_j)\|^2}{2\sigma^2})$, where $\sigma$ is a scaling parameter, typically set to the median of all pairwise distances. The degree matrix $D$ is then computed as a diagonal matrix where each diagonal element is the sum of the corresponding row in $A$:$D_{ii} = \sum_{j=1}^{n} A_{ij}$. Using these matrices, we define the normalized graph Laplacian $L$:$L = I - D^{-1/2}AD^{-1/2}$, where $I$ is the identity matrix. We then apply the diffusion kernel to this Laplacian: $K_t = \exp(-tL)$, where $K_t$ is the diffusion kernel at time $t$. The diffusion time $t$ is selected using the spectral gap method:$t_{opt} = \arg\max_t \left( \frac{d \log(\lambda_2(t))}{d \log(t)} \right)$, where $\lambda_2(t)$ is the second eigenvalue of $K_t$. The low-dimensional manifold representation $e(x_i)$ is then obtained by selecting the top $d$ eigenvectors of $K_{t_{opt}}$:$e(x_i) = [\phi_1(x_i), \phi_2(x_i), ..., \phi_d(x_i)]$, where $\phi_j$ are the eigenvectors of $K_{t_{opt}}$ corresponding to the $d$ largest eigenvalues.

In the second step, we perform NMF-based spectral clustering on these low-dimensional representations. Specifically, we construct a similarity matrix $S$ from the manifold representations:$S_{ij} = \exp(-\frac{\|e(x_i) - e(x_j)\|^2}{2\sigma^2})$. We then determine the optimal number of clusters $K$ by performing NMF with different values of $k$ and selecting the one that minimizes the Frobenius norm of the approximation error: $K = \arg\min_k \|S - W_k H_k^T\|_F$, where $W_k$ and $H_k$ are non-negative matrices resulting from NMF with $k$ components. Using this optimal $K$, we perform the final NMF: $\approx WH^T$, where $W$ and $H$ are non-negative matrices with $K$ columns. Finally, we assign each data point to a cluster based on the maximum value in each row of $W$:$c_i = \arg\max_j W_{ij}, \quad i = 1, ..., N$. This results in $K$ clusters $C = \{c_1, \ldots, c_K\}$, where the number of clusters $K$ is adaptively determined through the NMF process.

## 4.2 CAPABILITY DEGRADATION-AWARE RECOVERY INSTRUCTION SELECTION

**Capability Degradation Assessment** To prioritize the severely affected capabilities and finally achieve the balanced recovery of pruned LLMs, we need a measure of capability degradation to guide the data selection. For each cluster $c_k$ obtained in Section 4.1, we define the Capability degradation Score (CDS) with the Jensen-Shannon divergence (JSD) (Fuglede & Topsoe, 2004):

$$\text{CDS}(c_k) = \frac{1}{|c_k|} \sum_{(x,y)\in c_k} \frac{1}{|y|} \sum_{m=1}^{|y|} \text{JSD}(P(t_m|M_p, x)||P(t_m|M_o, x)). \tag{4}$$

Here, $P(t_m|M_p, x)$ represents the output probability distribution on the $m$-th token of the pruned model $M_p$ given input $x$. Taking an token $t_m^i$ in this distribution as an example, its corresponding probability is as follows:

$$P(t_m^i|M_p, x) = \frac{\exp(\frac{\text{logit}(t_m^i)}{\tau})}{\sum_{j=1}^{|\text{Voc}|} \exp(\frac{\text{logit}(t_m^j)}{\tau})}, \tag{5}$$

where $\tau$ is the softmax temperature and the $|\text{Voc}|$ indicates the vocabulary size. $\text{logit}(\cdot)$ is the logit value for tokens by LLM. Similarly, the $P(t_m|M_o, x)$ represents the output probability distribution for the original model $M_o$. The JSD is actualy the smoothed symmetrized and smoothed version of the Kullback–Leibler divergence (KLD) (Kullback & Leibler, 1951): $\text{JSD}(X||Y) = \frac{1}{2}\text{KLD}(X||M) + \frac{1}{2}\text{KLD}(Y||M)$. The distribution $M = \frac{1}{2}(X + Y)$ is the mixed distribution of $X$ and $Y$.

Thus, the obtained CDS quantifies the average performance degradation for data points in each capability cluster. The choice of JSD over simple loss variations as the performance degradation signal is motivated by its unique properties. First, its symmetry ensures consistent comparison between the pruned model $M_p$ and the original model $M_o$, while its bounded range (0 to 1) provides a normalized measure of divergence. This facilitates easier interpretation and comparison across different capability clusters. Moreover, JSD's robustness to outliers and its information-theoretic foundation allow for a more nuanced assessment of capability degradation, capturing subtle changes in model outputs that might not be apparent from loss or accuracy values alone (Dutta et al., 2024). The smoothing effect introduced by the use of the mixed distribution in JSD calculation also contributes to a more stable assessment across diverse instruction types. By employing JSD in our CDS calculation,

we obtain a comprehensive and reliable assessment of capability degradation, enabling more accurate identification and prioritization of the capabilities most severely affected by model pruning.

**Inter-capability Budget Allocation** Sampling a subset of high-quality data from the original set to achieve better training performance is the objective of the data selection process. To ensure the efficiency on data utilization and training process, the instruction data budget $B(B < N)$ should be maintained. Under this budget, we design an adaptive selection approach based on the above CDS for balanced recovery while focusing on severely affected capabilities. In detail, we allocate sampling budget to each cluster proportionally to its corresponding CDS:

$$n_k = \left\lfloor B \cdot \frac{\text{CDS}(c_k)}{\sum_{j=1}^{K} \text{CDS}(c_j)} \right\rfloor \tag{6}$$

where $n_k$ is the number of samples allocated to cluster $c_k$.

**Intra-capability Efficiency-Driven Sample Selection** To maximize computational efficiency during the recovery post-training phase, we need to select samples that offer the highest recovery benefit relative to their computational cost. Within each cluster $c_k$, we select top $n_k$ samples based on their Individual Efficiency Score (IES):

$$\text{IES}(x, y) = \frac{\frac{1}{|y|} \sum_{m=1}^{|y|} \text{JSD}(P(t_m|M_p, x) || P(t_m|M_o, x))}{\log \text{ComputationalCost}(x, y)}. \tag{7}$$

Here, ComputationalCost is instantiated with the quadratic term of sequence length $(|x| + |y|)^2$ as the approximated complexity for LLM training. The use of JSD captures the degree of divergence between the pruned and original models' outputs, indicating areas where capabilities have been most affected and thus offering the highest potential for targeted recovery. The logarithmic term balances the consideration of computational cost, allowing for a more nuanced selection that favors efficient samples without overly penalizing high-potential, moderately costly instances.

### 4.3 Negative Transfer Mitigation

To prevent performance degradation due to conflicting or irrelevant information, we need to detect and mitigate potential negative transfer. We introduce a Concept Consistency Graph (CCG) to model relationships between concepts in the selected data. Here, a concept refers to a key knowledge unit or semantic element extracted from an instruction tuning sample. Concepts play a crucial role in capturing the essential information within instructions and help in identifying potential conflicts that could lead to negative transfer. By managing relationships between concepts, we aim to maintain semantic consistency across the selected instruction tuning dataset, thereby reducing the risk of learning conflicting or irrelevant information during the recovery process.

**Definition 1** (Concept Consistency Graph). *A CCG is a graph $G = (V, E)$ where vertices $V$ represent concepts, and an edge $(v_i, v_j) \in E$ exists if concepts $v_i$ and $v_j$ co-occur in at least one instruction tuning sample without conflict.*

For each new sample $(x, y)$, we extract its concepts $C(x, y)$ and check for consistency:

$$\text{IsConsistent}(x, y) = \forall v_i, v_j \in C(x, y) : (v_i, v_j) \in E \text{ or } \{v_i, v_j\} \not\subset V \tag{8}$$

We only add samples that are consistent with the existing CCG, updating the graph with each addition. This approach ensures that we maintain a coherent set of instructions, minimizing the risk of negative transfer by avoiding the introduction of conflicting concepts during the recovery training process. The full version of our algorithm is provided in Algorithm 1.

### 4.4 Time Complexity Analysis

We provide a comprehensive analysis of the time complexity for the PASER algorithm.

**Theorem 1.** *The overall time complexity of PASER is $O(N \log N + NC^2)$, where $N$ is the number of instructions in $D$, and $C$ is the maximum number of concepts in any instruction tuning sample.*

---

**Algorithm 1** Post-training data Selection for efficient pruned large language model recovery (PASER)

---

**Input:** $M_o$: original model, $M_p$: pruned model, $D$: instruction tuning dataset, $B$: data budget, $U$: computational cost budget

    **procedure** PASER($M_o, M_p, D, B, U$)
        $C \leftarrow$ NMFSpectralClustering($\{e(x_i)|(x_i, y_i) \in D\}$)
        **for** $c_k \in C$ **do**
            CDS($c_k$) $\leftarrow$ ComputeCapabilityDegradationScore($c_k, M_o, M_p$)
        **end for**
        $\{n_k\} \leftarrow$ AllocateSamples($\{$CDS($c_k$)$\}, B$)
        $S \leftarrow \emptyset, G \leftarrow$ InitializeCCG()
        **for** $c_k \in C$ **do**
            $L_k \leftarrow$ SortByIESDescending($c_k$)
            $i \leftarrow 0, \text{count} \leftarrow 0$
            **while** count $< n_k$ and $i < |L_k|$ **do**
                $(x, y) \leftarrow L_k[i]$
                **if** IsConsistent($x, y, G$) and $\sum\limits_{(x',y') \in S \cup \{(x,y)\}}$ ComputationalCost($x', y'$) $\leq U$ **then**
                    $S \leftarrow S \cup \{(x, y)\}$
                    $G \leftarrow$ UpdateCCG($G, x, y$)
                    count $\leftarrow$ count $+ 1$
                **end if**
                $i \leftarrow i + 1$
            **end while**
        **end forreturn** $S$
    **end procedure**

---

*Proof.* We analyze each step of the algorithm in detail: The Semantic-Structural Recovery Instruction Clustering step, including SentenceBERT embedding, Diffusion Kernel computation, and NMF Spectral Clustering, has a dominant complexity of $O(N \log N)$. For the Capability Degradation Assessment step, computing JSD for each sample and calculating CDS for each cluster take $O(N)$ time in total. The Inter-capability Budget Allocation, which involves allocating the budget to clusters, has a time complexity of $O(K)$, where $K$ is the number of clusters. However, since $K \leq N$, this step does not dominate the overall complexity. During Intra-capability Efficiency-Driven Sample Selection, for each cluster $c_k$, we perform sorting by JSD ($O(|c_k| \log |c_k|)$), iterate through sorted samples ($O(|c_k|)$), perform consistency checks (IsConsistent, $O(C^2)$ per sample), and update the CCG ($O(C^2)$ per sample). Considering all clusters, this step's total complexity is $O(N \log N + NC^2)$. Thus, the overall time complexity is dominated by the clustering step and the intra-capability sample selection step. Therefore, the total time complexity is $O(N \log N + NC^2)$. □

In practice, $C$ is often much smaller than $N$ ($C \ll N$) and can be considered as a constant factor for large $N$. Thus, we can simplify the complexity to $O(N \log N)$. This analysis demonstrates that PASER is computationally efficient and scalable for large instruction tuning datasets.

## 5 EXPERIMENTS

### 5.1 EXPERIMENT SETUP

**Target LLMs:** The experiments are performed on several open-source popular English LLMs: LLaMA2-7B/13B/70B (Touvron et al., 2023) (hf version), LLaMA3-8B (Dubey et al., 2024)(instruct version), and bilingual LLMs: Baichuan2-7B/13B (Yang et al., 2023)(base version), which support both English and Chinese.

**Instruction Tuning Datasets:** As for the original recovery post-training data, we choose two different-size instruction tuning datasets: Alpaca (Taori et al., 2023) and LaMini (Wu et al., 2024). Alpaca contains 52K instruction-following samples generated using OpenAI's *text-davinci-003* model based on 175 human-written seed tasks. LaMini contains a total of 2.58M pairs of instructions and responses synthesized with *gpt-3.5-turbo* based on several existing resources of prompts, including

Table 1: Recovery performance of different instruction tuning data selection methods under various pruning schemes on LLaMA2-7B model. The 'bold' represents the best performance under the same pruning scheme. Here, the Alpaca is taken as the original dataset.

| Pruning | Recovery Post-training | WikiText2↓ | PTB↓ | BoolQ | PIQA | HellaSwag | WinoGrande | ARC-e | ARC-c | OBQA | Average |
|---|---|---|---|---|---|---|---|---|---|---|---|
| w/o pruning | w/o Training | 12.62 | 22.14 | 71.13 | 78.40 | 72.79 | 67.17 | 69.36 | 40.70 | 40.80 | 62.91 |
| LLM-Pruner ratio=25% | w/o Training | 20.34 | 38.81 | 61.87 | 76.61 | 65.86 | 60.22 | 63.13 | 37.37 | 39.40 | 57.78 |
| | Full Data | 736.42 | 1273.10 | 37.83 | 53.21 | 26.42 | 49.57 | 25.29 | 28.16 | 29.00 | 35.64 |
| | Random | 93.77 | 180.62 | 57.61 | 64.37 | 45.39 | 55.87 | 43.78 | 31.94 | 34.90 | 47.69 |
| | Instruction Mining | 23.31 | 40.63 | 61.59 | 75.68 | 66.08 | 60.71 | 62.34 | 37.96 | 39.20 | 57.65 |
| | IFD | 19.76 | 33.30 | 63.55 | 77.23 | 67.21 | 60.90 | 63.46 | 37.81 | 40.00 | 58.59 |
| | Nuggets | 20.02 | 35.19 | 63.62 | **77.43** | 67.36 | 61.08 | 63.77 | 37.64 | **39.90** | 58.69 |
| | PASER | **16.40** | **26.35** | **67.25** | 77.29 | **68.98** | **66.97** | **67.84** | **39.54** | 39.80 | **61.10** |
| SliceGPT ratio=25% | w/o Training | 44.53 | 80.07 | 65.54 | 66.87 | 54.16 | 63.38 | 58.46 | 34.56 | 36.90 | 54.27 |
| | Full Data | 38.24 | 68.53 | 68.75 | 69.84 | 57.92 | 66.18 | 62.37 | 36.82 | 38.30 | 57.17 |
| | Random | 41.86 | 74.92 | 66.89 | 68.21 | 55.79 | 64.56 | 60.23 | 35.47 | 37.60 | 55.54 |
| | Instruction Mining | 39.75 | 71.28 | 67.87 | 68.93 | 56.42 | 65.76 | 61.89 | 36.23 | 37.60 | 56.39 |
| | IFD | 37.75 | 67.48 | 69.23 | 70.54 | 58.38 | 67.12 | 63.75 | 37.18 | 38.40 | 57.80 |
| | Nuggets | 23.86 | 35.42 | 69.89 | 71.21 | 58.79 | 67.56 | **72.23** | 37.47 | 38.60 | 59.39 |
| | PASER | **12.24** | **21.53** | **72.75** | **79.84** | **73.92** | **69.18** | 71.37 | **41.82** | **41.30** | **64.31** |
| Wanda sparsity=2:4 | w/o Training | 42.10 | 76.85 | 69.30 | 71.99 | 53.06 | 62.75 | 60.94 | 28.07 | 34.60 | 54.39 |
| | Full Data | 27.63 | 50.22 | 70.77 | 74.87 | 63.78 | 65.26 | 65.30 | 34.04 | 37.10 | 58.73 |
| | Random | 35.98 | 65.24 | 69.68 | 73.14 | 58.65 | 63.69 | 63.16 | 31.91 | 36.20 | 56.63 |
| | Instruction Mining | 31.47 | 57.17 | 70.61 | 73.85 | 61.27 | 64.13 | 64.72 | 33.79 | 36.80 | 57.88 |
| | IFD | 25.82 | 46.78 | 71.06 | 75.57 | 64.15 | 65.38 | 66.55 | 35.63 | 37.60 | 59.42 |
| | Nuggets | 23.98 | 43.24 | **71.68** | 76.14 | 64.65 | 65.69 | 66.16 | 36.91 | 38.20 | 59.92 |
| | PASER | **14.13** | **27.22** | 70.77 | **77.87** | **71.78** | **66.26** | **68.30** | **39.04** | **40.10** | **62.02** |
| SparseGPT sparsity=50% | w/o Training | 19.26 | 36.41 | 71.22 | 75.60 | 62.85 | 66.06 | 69.11 | 36.86 | 37.80 | 59.93 |
| | Full Data | 25.83 | 47.26 | 69.10 | 74.15 | 59.68 | 67.76 | 63.74 | **39.59** | 37.80 | 58.83 |
| | Random | 28.74 | 50.85 | 67.84 | 75.39 | 57.14 | 68.92 | 59.76 | 37.34 | 36.60 | 57.57 |
| | Instruction Mining | 24.08 | 45.51 | 70.50 | 74.47 | 61.91 | 65.40 | 67.73 | 36.49 | 37.40 | 59.13 |
| | IFD | 21.19 | 40.05 | 71.06 | 75.13 | 62.79 | 65.72 | 68.80 | 36.23 | 37.40 | 59.56 |
| | Nuggets | 16.21 | 28.95 | 71.64 | 75.67 | 63.33 | 66.05 | 69.49 | 36.60 | 37.40 | 60.03 |
| | PASER | **13.33** | **23.77** | **74.79** | 78.38 | **66.62** | **69.03** | **72.57** | 38.70 | **39.40** | **62.78** |

self-instruct (Wang et al., 2023), P3 (Sanh et al., 2022), FLAN (Longpre et al., 2023) and Alpaca (Taori et al., 2023).

**Base Pruning Schemes:** Different pruning schemes are incorporated into our experiments to explore the applicability of PASER, ranging from structured pruning methods: LLM-Pruner (Ma et al., 2023), SliceGPT (Ashkboos et al., 2024), semi-structured method: Wanda (Sun et al., 2024), and unstructured pruning methods: SparseGPT (Frantar & Alistarh, 2023).

**Instruction Tuning Data Selection Baselines:** In addition to the random data selection, we also compare PASER with several recent general instruction tuning data selection baselines: Instruction Mining (Cao et al.), IFD (Li et al., 2024a), Nuggets (Li et al., 2024b). Note none of these baselines are customized for post-pruning recovery training scenario. Besides, the evaluation performance of recovery training with the full original dataset is also compared.

**Evaluation Datasets and Tasks** To comprehensively evaluate the performance of recovered LLMs, we employ seven common sense reasoning datasets:BoolQ (Clark et al., 2019), PIQA (Bisk et al., 2020), HellaSwag (Zellers et al., 2019), WinoGrande (Sakaguchi et al., 2021), ARC-Easy (Clark et al., 2018), ARC-Challenge (Clark et al., 2018), and OpenbookQA (Mihaylov et al., 2018). In the practice, we relies on the open-source library[1] to implement the evaluation, where the model needs to rank the choices in the multiple choice tasks or generate the answer in the open-ended generation tasks. The whole process is conducted in the *zero-shot* manner. Besides, we follow (Ma et al., 2023) to evaluate the language modeling capability with the zero-shot perplexity (PPL) analysis on WikiText2 (Merity et al., 2022) and PTB (Marcus et al., 1993).

More experimental details on the implementation are provided in the appendix A.

## 5.2 RECOVERY PERFORMANCE FOR DIFFERENT PRUNING SCHEMES

We evaluate the recovery performance of LLaMA2-7B using different instruction tuning data selection methods under structured pruning, semi-structured pruning, and unstructured pruning, respectively.

---

[1] https://github.com/EleutherAI/lm-evaluation-harness

According to the results in Table 1, directly employing full data can indeed bring the sub-optimal recovery performance, especially under the LLM-Pruner. This is because the full version of data contains some irrelevant or conflicting information for capability recovery, resulting in the negative transfer during the training phase. Meanwhile, even the general instruction tuning data selection methods like IFD and Nuggets can bring better reasoning and language model performance than full data and random in most cases, validating the necessity of conducting recovery data selection. Furthermore, we can find that previous selection methods can hardly help model recover to the level of unpruned status, under the limited data budget. However, our PASER can not only outperform baselines, but also reduce the averaged reasoning performance degradation to less than 3% under LLM-Pruner, Wanda, and SparseGPT. Especially, when pruning LLaMA2-7B with SliceGPT, our PARSER can improve the average reasoning performance to 64.31, higher than the unpruned model. Besides, its zero-shot perplexity on WikiText2 and PTB is also lower than unpruned model slightly. This suggests that allocating recovery budget according to capability degradation levels and prioritizing most-affected samples exhibit the potential of help pruned LLM recover to the capability level of unpruned status. Besides, PASER can also be extended to other LLM post-compression scenarios, like the post-quantization recovery. The corresponding results and analysis are provided in Appendix B.

## 5.3 Recovery Performance with Different Instruction Tuning Datasets

In addition to the recovery performance on Alpaca shown in Table 1, we also explore the corresponding results on another larger dataset, LaMini. Especially, considering the space limitation and more severe performance degradation of structured pruning schemes, we provide the experiments results on LLM-Pruner and SliceGPT on Table 2. From this table, we can observe that PASER can still consistently outperform other data selection and random methods. Besides, comparing the results in Table 1 and 2, it can be found that improving the data scale (from 10K to 10K samples) indeed facilitates the recovery performance though the significantly increased computational overhead, which is consistent with the Scaling Law (Kaplan et al., 2020). We can also notice that the performance of full data on LaMini is relatively competitive, which is because the proportion of conflicting or negative data for recovery is much lower than that in Alpaca.

Table 2: Recovery performance of different instruction tuning data selection methods under two structured pruning schemes on LLaMA2-7B model. The 'bold' represents the best performance under the same pruning scheme. Here, the LaMini is taken as the original dataset.

| Pruning | Recovery Post-training | WikiText2↓ | PTB↓ | BoolQ | PIQA | HellaSwag | WinoGrande | ARC-e | ARC-c | OBQA | Average |
|---|---|---|---|---|---|---|---|---|---|---|---|
| w/o pruning | w/o Training | 12.62 | 22.14 | 71.13 | 78.40 | 72.79 | 67.17 | 69.36 | 40.70 | 40.80 | 62.91 |
| LLM-Pruner ratio=25% | w/o Training | 20.34 | 38.81 | 61.87 | 76.61 | 65.86 | 60.22 | 63.13 | 37.37 | 39.40 | 57.78 |
| | Full Data | 16.28 | 27.12 | 76.79 | 77.26 | 66.93 | 65.75 | **72.18** | **40.44** | 39.40 | 62.68 |
| | Random | 18.40 | 32.15 | 70.52 | 77.21 | 67.42 | 64.89 | 69.41 | 38.14 | 38.90 | 60.93 |
| | Instruction Mining | 17.83 | 28.87 | 69.62 | 77.45 | 67.26 | 65.34 | 69.14 | 37.71 | 38.80 | 60.76 |
| | IFD | 18.54 | 31.23 | 71.29 | 77.21 | 67.05 | 64.88 | 68.26 | 37.23 | 38.60 | 60.65 |
| | Nuggets | 18.27 | 30.90 | 69.38 | 77.35 | 67.91 | 63.59 | 70.11 | 39.02 | 39.60 | 60.99 |
| | PASER | **13.45** | **22.63** | **77.98** | **78.36** | **69.93** | **67.31** | 72.05 | 40.28 | **40.60** | **63.79** |
| SliceGPT ratio=25% | w/o Training | 44.53 | 80.07 | 65.54 | 66.87 | 54.16 | 63.38 | 58.46 | 34.56 | 36.90 | 54.27 |
| | Full Data | 24.36 | 35.64 | 70.75 | 70.84 | 58.92 | 67.18 | 64.37 | 37.82 | 38.30 | 58.31 |
| | Random | 39.86 | 70.92 | 68.89 | 69.21 | 56.79 | 65.56 | 62.23 | 36.47 | 37.60 | 56.68 |
| | Instruction Mining | 37.75 | 67.28 | 69.87 | 69.93 | 57.42 | 66.76 | 63.89 | 37.23 | 37.60 | 57.53 |
| | IFD | 25.75 | 53.48 | 71.23 | 71.54 | 59.38 | 68.12 | 65.75 | 38.18 | 38.40 | 58.94 |
| | Nuggets | 21.86 | 31.42 | 71.89 | 72.21 | 59.79 | 68.56 | **74.23** | 38.47 | **41.60** | 60.96 |
| | PASER | **14.27** | **23.53** | **74.75** | **81.84** | **75.92** | **70.18** | 73.37 | **42.82** | 41.30 | **65.74** |

## 5.4 Robustness over Various Target Large Language Models

To validate whether PASER can maintain the robust effectiveness among various target LLMs, we conduct the experiments on LLaMA2-7B/13B/70B, LLaMA3-8B, Baichuan2-7B/13B, under LLM-Pruner. According to results in Table 3, we can first observe the model capability under high pruning ratio (50%) is hard to recover to unpruned level, especially for relatively smaller model like LLaMA2-13B and Baichuan2-13B. Though, PASER can still outperform random and best-performing data selection baseline, Nuggets, by 4.41 and 2.31 points, respectively on average. Especially, for LLaMA2-70B, our PARSER can control the averaged reasoning performance degradation to less than 3%. This can be explained that the structure redundancy in 70B model is relatively higher, paving

Table 3: Recovery performance of different instruction tuning data selection methods on different target LLMs under LLM-Pruner scheme. The 'bold' represents the best performance on the same target LLM. Here, the Alpaca is taken as the original dataset.

| Target LLM | Recovery Post-training | WikiText2↓ | PTB↓ | BoolQ | PIQA | HellaSwag | WinoGrande | ARC-e | ARC-c | OBQA | Average |
|---|---|---|---|---|---|---|---|---|---|---|---|
| LLaMA2-13B ratio=50% | w/o pruning | 11.58 | 20.24 | 69.02 | 78.73 | 76.60 | 69.69 | 73.23 | 44.20 | 42.00 | 64.78 |
| | w/o Training | 73.52 | 151.19 | 48.99 | 69.10 | 53.03 | 53.12 | 46.80 | 31.40 | 39.60 | 48.86 |
| | Full Data | 27.74 | 45.08 | 63.52 | 74.21 | 62.28 | 56.43 | 62.79 | 35.75 | 39.80 | 56.40 |
| | Random | 39.85 | 76.20 | 59.28 | 73.71 | 60.02 | 56.09 | 59.81 | 34.04 | 39.40 | 54.62 |
| | Instruction Mining | 44.37 | 80.82 | 57.25 | 73.22 | 59.02 | 57.28 | 62.34 | 34.64 | 34.90 | 54.09 |
| | IFD | 38.61 | 73.25 | 57.63 | 72.89 | 59.20 | 57.81 | 62.19 | 34.90 | 38.80 | 54.77 |
| | Nuggets | 33.50 | 61.26 | 61.08 | 74.24 | 60.98 | 55.39 | 60.96 | 34.90 | 39.20 | 55.25 |
| | **PASER** | **21.67** | **35.09** | **65.42** | **75.69** | **63.24** | **58.56** | **63.35** | **36.45** | **40.60** | **57.62** |
| LLaMA2-70B ratio=50% | w/o pruning | 8.92 | 15.59 | 81.25 | 79.98 | 82.45 | 76.56 | 80.85 | 54.74 | 46.20 | 71.72 |
| | w/o Training | 46.81 | 92.36 | 68.43 | 72.14 | 66.87 | 65.43 | 67.32 | 45.18 | 42.60 | 61.14 |
| | Full Data | 31.76 | 56.83 | 74.89 | 74.62 | 74.93 | 69.84 | 74.41 | 48.45 | 41.80 | 65.56 |
| | Random | 25.34 | 48.72 | 72.56 | 73.83 | 72.21 | 68.18 | 72.38 | 47.27 | 41.80 | 64.03 |
| | Instruction Mining | 23.16 | 43.45 | 74.13 | 76.39 | 74.38 | 70.75 | 76.19 | 48.92 | 44.40 | 66.74 |
| | IFD | 22.87 | 43.68 | 75.32 | **78.50** | 75.04 | 71.23 | 75.97 | 50.61 | 44.20 | 67.27 |
| | Nuggets | 19.63 | 36.24 | 76.18 | 77.18 | 76.28 | 71.89 | 76.80 | 51.20 | 44.60 | 67.73 |
| | **PASER** | **12.35** | **21.82** | **78.81** | 78.38 | **79.15** | **74.25** | **78.42** | **52.95** | **45.40** | **69.62** |
| LLaMA3-8B ratio=25% | w/o pruning | 7.36 | 12.87 | 83.09 | 78.62 | 75.80 | 72.06 | 81.65 | 56.74 | 43.00 | 70.14 |
| | w/o Training | 15.47 | 28.31 | 74.78 | 72.95 | 67.22 | 65.85 | 73.49 | 50.07 | 39.80 | 63.45 |
| | Full Data | 9.58 | 16.73 | 80.60 | 76.87 | 72.91 | 69.82 | 78.38 | 54.47 | 41.80 | 67.84 |
| | Random | 12.52 | 22.17 | 77.69 | 74.91 | 70.06 | 67.73 | 75.93 | 52.20 | 40.80 | 65.60 |
| | Instruction Mining | 13.25 | 23.47 | 76.85 | 74.69 | 69.74 | 67.52 | 76.34 | 52.55 | 40.60 | 65.47 |
| | IFD | 11.04 | 19.31 | 78.94 | 75.69 | 71.25 | 68.46 | 77.57 | 53.34 | 41.20 | 66.64 |
| | Nuggets | 10.31 | 18.02 | 79.77 | 76.28 | 72.01 | 69.18 | 78.38 | 53.90 | 41.60 | 67.30 |
| | **PASER** | **8.09** | **14.16** | **82.43** | **78.44** | **75.29** | **71.62** | **81.02** | **56.61** | **43.40** | **69.83** |
| Baichuan2-7B ratio=25% | w/o pruning | 14.42 | 26.78 | 74.04 | 77.26 | 72.24 | 68.43 | 75.08 | 42.49 | 39.80 | 64.19 |
| | w/o Training | 28.30 | 53.34 | 49.76 | 73.29 | 63.85 | 60.14 | 67.00 | **40.27** | **40.00** | 56.33 |
| | Full Data | 25.29 | 35.81 | 52.20 | 74.81 | 66.24 | 62.75 | 67.72 | 39.42 | 38.60 | 57.39 |
| | Random | 27.04 | 46.83 | 53.15 | 73.94 | 64.60 | 60.62 | 68.73 | 39.39 | 39.20 | 57.09 |
| | Instruction Mining | 34.24 | 60.93 | 71.39 | 69.56 | 55.65 | 61.74 | 57.13 | 33.90 | 34.10 | 54.78 |
| | IFD | 24.83 | 37.81 | 53.49 | 73.86 | 64.69 | 61.25 | 68.77 | 40.24 | 39.20 | 57.36 |
| | Nuggets | 21.48 | 37.65 | 56.25 | 74.27 | **66.41** | 62.73 | 68.73 | 38.29 | 38.20 | 57.84 |
| | **PASER** | **16.92** | **30.76** | **66.95** | **74.82** | 65.19 | **63.38** | **69.69** | 38.98 | 39.00 | **59.70** |
| Baichuan2-13B ratio=50% | w/o pruning | 11.23 | 18.04 | 79.60 | 77.31 | 75.32 | 70.09 | 77.36 | 47.10 | 44.00 | 67.25 |
| | w/o Training | 58.41 | 116.26 | 65.27 | 69.58 | 62.91 | 60.98 | 64.24 | 39.55 | 40.60 | 57.59 |
| | Full Data | 24.35 | 42.68 | 71.84 | **72.15** | 68.73 | 63.75 | 70.67 | **42.91** | 41.40 | 61.64 |
| | Random | 40.44 | 76.57 | 67.06 | 70.23 | 65.72 | 61.29 | 68.33 | 41.02 | 40.20 | 59.12 |
| | Instruction Mining | 36.82 | 70.45 | 68.26 | 70.60 | 66.47 | 61.64 | 67.56 | 40.55 | 40.60 | 59.38 |
| | IFD | 33.45 | 63.23 | 69.45 | 71.38 | 67.38 | 62.55 | 69.12 | 41.49 | 41.00 | 60.34 |
| | Nuggets | 28.96 | 53.31 | 70.64 | 71.99 | 68.48 | 63.30 | 69.89 | 41.96 | 41.40 | 61.09 |
| | **PASER** | **14.62** | **29.82** | **75.83** | 71.76 | **72.18** | **67.24** | **73.95** | 42.52 | **42.80** | **63.75** |

the way for effective recovery through data selection under high pruning ratios. As for the second smallest model, LLaMA3-8B, PARSER can recover the reasoning performance to the 99.56% of the unpruned status, which further demonstrates the robustness of PARSER over different target LLMs. Finally, the performance of various recovery methods including PARSER on Baichuan2-7B is not satisfying enough given only 25% pruning ratio, which can be attributed to the fact that the pruning process has severely damaged the model internal structure.

## 5.5 ABLATION STUDY

To validate the contribution of each component in PASER, we conduct comprehensive ablation studies. Specifically, we evaluate three variants: (1) PASER w/o $S^2$RIC: replacing semantic-structural clustering with random clustering while keeping other modules unchanged; (2) PASER w/o CDARIS: randomly sampling equal number of instructions from each cluster instead of using capability degradation-aware selection; (3) PASER w/o NTM: removing the negative transfer mitigation module.

The results in Table 4 demonstrate that all three components contribute positively to model recovery across different pruning schemes. The semantic-structural clustering effectively identifies capability-specific instruction groups, leading to 0.18-1.43 points improvement in reasoning performance. Its removal causes degradation in both language modeling (increased perplexity) and reasoning tasks, particularly evident under structured pruning schemes like LLM-Pruner and SliceGPT. The capability degradation-aware selection mechanism enhances recovery efficiency through adaptive budget allocation, contributing 0.29-1.63 points improvement in reasoning tasks while maintaining

Table 4: Ablation study results on LLaMA2-7B for each component under different pruning schemes. The "Reason" indicates the averaged performance on 7 reasoning datasets.

| Ablation Variant | LLM-Pruner (25%) | | | SliceGPT (25%) | | | Wanda (2:4) | | | SparseGPT (50%) | | |
|---|---|---|---|---|---|---|---|---|---|---|---|---|
| | WikiText2↓ | PTB↓ | Reason↑ | WikiText2↓ | PTB↓ | Reason↑ | WikiText2↓ | PTB↓ | Reason↑ | WikiText2↓ | PTB↓ | Reason↑ |
| w/o S²RIC | 18.73 | 32.84 | 59.67 | 14.83 | 25.42 | 63.03 | 15.84 | 30.25 | 61.19 | 14.89 | 26.31 | 62.60 |
| w/o CDARIS | 17.56 | 30.15 | 60.26 | 14.16 | 24.92 | 62.68 | 15.46 | 29.48 | 61.23 | 14.62 | 25.84 | 62.49 |
| w/o NTM | 19.82 | 35.60 | 59.25 | 15.37 | 27.81 | 61.92 | 16.79 | 31.52 | 61.34 | 15.91 | 28.19 | 61.76 |
| PASER | 16.40 | 26.35 | 61.10 | 12.24 | 21.53 | 64.31 | 14.13 | 27.22 | 62.02 | 13.33 | 23.77 | 62.78 |

stable language modeling performance. Notably, negative transfer mitigation shows significant impact (0.68-2.39 points improvement), especially under high pruning ratios, highlighting its importance in preventing conflicting information during recovery training. These improvements are consistently observed across different pruning schemes, with particularly pronounced effects in structured pruning where capability degradation tends to be more severe and uneven.

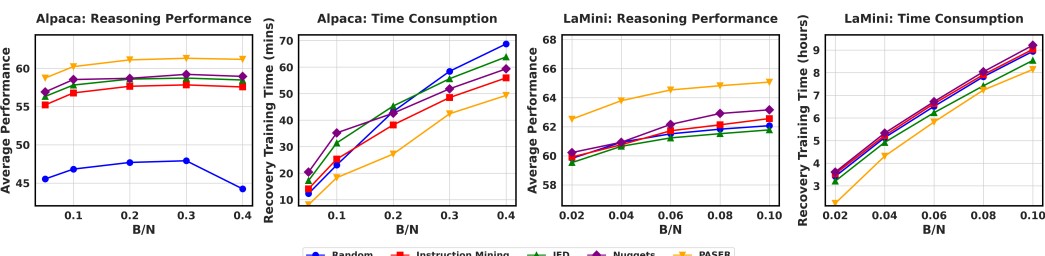

Figure 2: Average reasoning performance and recovery post-training time consumption curves corresponding to different instruction tuning data selection methods.

### 5.6 RECOVERY POST-TRAINING EFFICIENCY ANALYSIS

To highlight PARSER's advantages on recovery post-training efficiency, we conduct the experiments under different data budgets $B$ and different datasets and record the corresponding averaged reasoning performance and training time in Figure 2. From the first and third subfigures, we can observe that PARSER can obtain best recovery performance under different $B/N$ on Alpaca and LaMini. Interestingly, in the first subfigure, when rising $B/N$ from 0.3 to 0.4, the reasoning performance of Random even decreases. It is because expanding data scale also introduces the conflicting or negative data existing in the original dataset. From the second and fourth subfigures, PARSER consistently consumes the least training time, which can be attributed to the efficiency-driven sample selection process in PARSER. This advantage can be more obvious under low $B/N$ like 0.02 on LaMini. This is because increasing data budget will force PARSER to select some relatively more time-consuming samples given the fixed original dataset, weakening its efficiency superiority.

In addition, we also discuss the performance under knowledge distillation in Appendix C, provide the detailed ablation study results in Appendix D, explore other possible clustering methods in Appendix E, evaluate on mathematical tasks F and provide case study in Appendix G and H.

## 6 CONCLUSION AND FUTURE WORKS

Recovery post-training has been an important procedure after large language model pruning to restore the critical capabilities. Previous works directly utilize the full instruction tuning dataset, facing high computation cost and risks of untargeted recovery and negative transfer. In this work, we propose the post-training data selection method for efficient pruned model recovery. According to capability degradation degrees, we allocate selection budget among different capability data obtained through semantic-structural clustering. We then select samples where model behavior has been severely affected while considering computation cost, and introduce a concept consistency graph to mitigate negative transfer. Extensive experiments on different LLMs and pruning schemes demonstrate the effectiveness of our framework. Future work will explore other optimization approaches like data augmentation and scheduling to further improve recovery efficiency.

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

## A    IMPLEMENTATION DETAILS

Most of experiments are conducted on the server with $8 \times$ RTX 6000 Ada GPUs. During the recovery post-training phase, we take the the low-rank approximation, LoRA (Hu et al.), to improve the efficiency. The corresponding hyperparameters are set as following: rank=8, batch size=64, epochs=2, learning rate = 1e-4 (Alpaca series experiments), 5e-5 (LaMini series experiments). As for the structured pruning, we set the pruning ratio as 25% for LLaMA2-7B/LLaMA3-8B/Baichuan2-7B and 50% for LLaMA2-13B/LLaMA2-70B/Baichuan-13B models. For the other two kinds of pruning schemes, we follow the previous work (Frantar & Alistarh, 2023); Specifically, we adopt the 2:4 semi-structured sparsity patterns and implement 50% unstructured weight sparsity. Except the experiments in Section 5.6, we set the ratio of recovery data budget $B$ to original dataset size $N$ as 20% for Alpaca and 4% for LaMini. As for the implementation of concept extraction in Section 4.3, we use the open-source library `rake-nltk` [2]. To ensure statistical robustness, all the results reported in this paper are the averages of five runs with different seeds. Statistical significance is also assessed using two-tailed independent t-tests, with results considered significant when $p < 0.01$.

Table 5: Recovery performance of different instruction tuning data selection methods under various LLM quantization schemes on LLaMA2-7B model. The 'bold' represents the best performance under the same quantization scheme. Here, the Alpaca is taken as the original dataset.

| Quantization | Recovery Post-training | WikiText2↓ | PTB↓ | BoolQ | PIQA | HellaSwag | WinoGrande | ARC-e | ARC-c | OBQA | Average |
|---|---|---|---|---|---|---|---|---|---|---|---|
| w/o Quant | w/o training | 12.62 | 22.14 | 71.13 | 78.40 | 72.79 | 67.17 | 69.36 | 40.70 | 40.80 | 62.91 |
| RTN 4 bits | w/o training | 18.14 | 33.28 | 66.52 | 74.95 | 69.24 | 63.91 | 65.58 | 38.07 | 35.10 | 59.05 |
| | Full Data | 15.83 | 27.41 | 67.35 | 75.70 | 69.94 | 64.57 | 66.22 | 38.48 | 35.90 | 59.74 |
| | Random | 16.72 | 29.56 | 64.53 | 73.08 | 67.48 | 62.28 | 63.93 | 37.13 | 33.90 | 57.48 |
| | Instruction Mining | 16.05 | 27.83 | 66.73 | 75.15 | 69.43 | 64.08 | 65.74 | 38.18 | 35.30 | 59.23 |
| | IFD | 15.21 | 25.74 | 68.16 | 76.40 | 70.60 | 65.18 | 66.83 | 38.83 | 37.40 | 60.49 |
| | Nuggets | 14.68 | 24.53 | 68.99 | 77.13 | 71.28 | 65.82 | 67.46 | 39.21 | 38.70 | 61.23 |
| | PASER | **14.21** | **23.37** | **70.43** | **78.41** | **72.47** | **66.92** | **68.54** | **39.81** | **41.50** | **62.58** |
| GPTQ 4 bits | w/o training | 15.96 | 27.86 | 67.82 | 76.15 | 70.35 | 64.95 | 66.59 | 38.69 | 36.90 | 60.21 |
| | Full Data | 15.62 | 26.95 | 68.00 | 76.31 | 70.50 | 65.09 | 66.73 | 38.78 | 37.40 | 60.40 |
| | Random | 16.31 | 28.74 | 66.81 | 75.24 | 69.49 | 64.14 | 65.79 | 38.22 | 35.70 | 59.34 |
| | Instruction Mining | 15.37 | 26.42 | 68.31 | 76.58 | 70.75 | 65.33 | 66.96 | 38.93 | 37.90 | 60.68 |
| | IFD | 14.83 | 25.16 | 68.96 | 77.15 | 71.29 | 65.83 | 67.47 | **40.21** | 39.00 | 61.42 |
| | Nuggets | 13.52 | 22.93 | 69.74 | 77.83 | 71.93 | 66.43 | 68.06 | 39.56 | 40.20 | 61.96 |
| | PASER | **12.95** | **21.84** | **71.20** | **79.12** | **73.12** | **67.53** | **69.14** | 40.18 | **42.90** | **63.31** |

## B    EXTENDED EXPERIMENTS ON POST-QUANTIZATION RECOVERY TRAINING

Though the method descriptions and the experiments in the main body are mainly around the LLM pruning scenario, our PASER framework can actually be extended seamlessly to other LLM compression scenario. To further demonstrate its applicability, we conduct the experiments on post-quantization recovery training and compare our PASER with corresponding instruction tuning data selection baselines. In detail, we choose two most representative methods: Round-To-Nearest (RTN) (Frantar & Alistarh, 2022; Yao et al., 2022), GPTQ (Frantar et al., 2023), to perform the LLM quantization. It should be clarified that RTN method, which rounds all weights to the nearest quantized value on exactly the same asymmetric per-row grid, is actually the fundamental technique in most works about LLM quantization (Frantar & Alistarh, 2022; Yao et al., 2022; Park et al.). Its runtimes scales well to the models with many billion parameters due to the direct rounding. According to the results provided in Table 5 and 6, we can observe that the PASER can still effectively enhance the recovery performance and outperform the data selection baselines on averaged reasoning performance and zero-shot perplexity for both LLaMA2-7B and LLaMA2-13B models. Meanwhile, recovery data selection baselines can indeed achieve the stronger performance than full data and random baselines, which validates the necessity of conducting recovery data selection even in the LLM quantization scenario. Furthermore, comparing these results with Table 1 and 3, it can be noticed that the improvement space of PASER in Table 5 and 6 has been reduced to some extent. This is because the post-compression performance of such quantization schemes has been competitive enough, which can reflected from the w/o training row.

---

[2]https://pypi.org/project/rake-nltk/

Table 6: Recovery performance of different instruction tuning data selection methods under various LLM quantization schemes on LLaMA2-13B model. The 'bold' represents the best performance under the same quantization scheme. Here, the Alpaca is taken as the original dataset.

| Quantization | Recovery Post-training | WikiText2↓ | PTB↓ | BoolQ | PIQA | HellaSwag | WinoGrande | ARC-e | ARC-c | OBQA | Average |
|---|---|---|---|---|---|---|---|---|---|---|---|
| w/o Quant | w/o training | 11.58 | 20.24 | 69.02 | 78.73 | 76.60 | 69.69 | 73.23 | 44.20 | 42.00 | 64.78 |
| RTN 4 bits | w/o training | 17.53 | 32.34 | 63.15 | 74.59 | 72.62 | 65.94 | 69.17 | 41.49 | 37.00 | 60.57 |
| | Full Data | 16.95 | 31.02 | 63.59 | 75.02 | 73.04 | 66.33 | 69.58 | 41.75 | 37.50 | 60.97 |
| | Random | 17.86 | 33.15 | 62.00 | 73.48 | 71.55 | 64.94 | 68.13 | 40.84 | 35.20 | 59.45 |
| | Instruction Mining | 17.24 | 31.68 | 62.83 | 74.27 | 72.32 | 65.65 | 68.87 | 41.29 | 36.10 | 60.19 |
| | IFD | 15.63 | 28.39 | 65.03 | 76.37 | 74.34 | 67.49 | 70.80 | 42.46 | 39.60 | 62.30 |
| | Nuggets | 15.08 | 27.15 | 65.45 | 76.76 | **76.72** | 67.84 | 71.17 | 42.70 | 40.20 | 62.98 |
| | PASER | **12.34** | **23.08** | **67.33** | **78.50** | 76.41 | **69.37** | **72.78** | **43.67** | **41.70** | **64.25** |
| GPTQ 4 bits | w/o training | 14.74 | 26.86 | 64.68 | 76.04 | 74.02 | 67.20 | 70.49 | 42.28 | 39.10 | 61.97 |
| | Full Data | 16.02 | 29.34 | 63.62 | 75.05 | 73.07 | 66.35 | 69.61 | 41.76 | 37.40 | 60.98 |
| | Random | 14.58 | 26.52 | 64.82 | 76.17 | 74.15 | 67.32 | 70.61 | 42.36 | 39.30 | 62.10 |
| | Instruction Mining | 13.67 | 24.59 | 66.37 | 77.58 | 75.51 | 68.56 | 71.91 | **44.15** | 41.30 | 63.63 |
| | IFD | 13.51 | 24.26 | **68.46** | 77.66 | 75.59 | 68.63 | 71.99 | 43.20 | 41.40 | 63.85 |
| | Nuggets | 12.76 | 22.92 | 67.21 | 78.34 | 76.25 | 69.24 | 72.63 | 43.59 | 42.70 | 64.28 |
| | PASER | **11.25** | **20.93** | 68.11 | **79.16** | **77.05** | **69.97** | **73.39** | 44.04 | **44.20** | **65.13** |

Table 7: Knowledge distillation recovery performance of different instruction tuning data selection methods under various pruning schemes on LLaMA2-7B model. The 'bold' represents the best performance under the same pruning scheme. Here, the Alpaca is taken as the original dataset.

| Pruning | Recovery Post-training | WikiText2↓ | PTB↓ | BoolQ | PIQA | HellaSwag | WinoGrande | ARC-e | ARC-c | OBQA | Average |
|---|---|---|---|---|---|---|---|---|---|---|---|
| w/o pruning | w/o Training | 12.62 | 22.14 | 71.13 | 78.40 | 72.79 | 67.17 | 69.36 | 40.70 | 40.80 | 62.91 |
| LLM-Pruner ratio=25% | w/o Training | 20.34 | 38.81 | 61.87 | 76.61 | 65.86 | 60.22 | 63.13 | 37.37 | 39.40 | 57.78 |
| | Full Data | 24.72 | 43.91 | 63.30 | 76.01 | 67.18 | 62.27 | 64.23 | 36.86 | 39.20 | 58.44 |
| | Random | 23.82 | 41.20 | **68.03** | 74.89 | 66.27 | 64.51 | 64.65 | 32.58 | 38.30 | 58.46 |
| | Instruction Mining | 22.65 | 39.40 | 62.17 | 75.98 | 66.74 | 61.29 | 63.01 | 38.32 | 39.60 | 58.16 |
| | IFD | 19.17 | 32.30 | 64.13 | 77.55 | 67.89 | 61.56 | 64.09 | 38.19 | **40.40** | 59.12 |
| | Nuggets | 18.64 | 32.19 | 64.46 | 76.66 | 67.26 | 64.88 | 66.50 | 36.52 | 39.20 | 59.35 |
| | PARSER | **15.91** | **25.39** | 67.89 | **77.81** | 69.62 | **67.63** | 68.46 | **39.87** | 40.20 | **61.64** |
| SliceGPT ratio=25% | w/o training | 44.53 | 80.07 | 65.54 | 66.87 | 54.16 | 63.38 | 58.46 | 34.56 | 36.90 | 54.27 |
| | Full Data | 35.48 | 66.25 | 69.35 | 70.34 | 58.50 | 66.76 | 62.95 | 37.14 | 38.70 | 57.68 |
| | Random | 38.63 | 65.67 | 67.19 | 68.59 | 56.21 | 64.94 | 60.63 | 35.61 | 37.80 | 55.85 |
| | Instruction Mining | 35.56 | 62.14 | 68.41 | 69.51 | 57.08 | 66.33 | 62.51 | 36.59 | 38.00 | 56.92 |
| | IFD | 33.50 | 61.33 | 69.51 | 70.82 | 58.70 | 67.49 | 64.09 | 37.22 | 38.50 | 58.05 |
| | Nuggets | 21.39 | 32.83 | 70.17 | 71.49 | 59.11 | 67.94 | **72.51** | 37.54 | 38.70 | 59.64 |
| | PASER | **11.87** | **20.91** | **73.43** | **80.32** | **74.46** | **69.76** | 71.95 | **42.26** | **41.70** | **64.84** |
| Wanda sparsity=2:4 | w/o training | 42.10 | 76.85 | 69.30 | 71.99 | 53.06 | 62.75 | 60.94 | 28.07 | 34.60 | 54.39 |
| | Full Data | 25.92 | 47.85 | 71.09 | 75.14 | 64.10 | 65.62 | 65.64 | 34.38 | 37.50 | 59.07 |
| | Random | 34.98 | 63.47 | 70.18 | 73.62 | 59.15 | 63.83 | 63.70 | 32.13 | 36.50 | 57.02 |
| | Instruction Mining | 30.56 | 55.56 | 71.03 | 73.97 | 61.69 | 64.56 | 64.86 | 33.93 | 37.00 | 58.15 |
| | IFD | 24.08 | 41.44 | 71.78 | 75.89 | 64.83 | 65.72 | **68.89** | 35.97 | 38.00 | 60.15 |
| | Nuggets | 23.14 | 40.10 | **72.26** | 76.50 | 65.33 | 66.03 | 66.52 | 37.27 | 38.60 | 60.36 |
| | PASER | **13.84** | **23.54** | 71.25 | **78.15** | **72.06** | **66.64** | 68.66 | **39.38** | **40.50** | **62.38** |
| SparseGPT sparsity=50% | w/o training | 19.26 | 36.41 | 71.22 | 75.60 | 62.85 | 66.06 | 69.11 | 36.86 | 37.80 | 59.93 |
| | Full Data | 28.17 | 52.82 | 68.52 | 75.77 | 57.84 | 69.26 | 60.43 | 37.72 | 37.00 | 58.08 |
| | Random | 25.31 | 43.22 | 69.74 | 74.91 | 60.28 | 68.10 | 64.06 | **39.95** | **39.80** | 59.55 |
| | Instruction Mining | 21.56 | 39.61 | 71.12 | 74.85 | 62.53 | 66.06 | 68.07 | 36.85 | 37.80 | 59.61 |
| | IFD | 17.76 | 31.25 | 71.70 | 75.76 | 63.43 | 66.06 | 69.14 | 36.59 | 37.60 | 60.04 |
| | Nuggets | 14.83 | 25.38 | 72.18 | 75.95 | 63.91 | 66.29 | 69.75 | 36.86 | 37.70 | 60.38 |
| | PASER | **13.00** | **22.24** | **75.07** | **78.66** | **66.90** | **69.31** | 72.85 | 38.89 | 39.60 | **63.04** |

# C    EXTENDED EXPERIMENTS ON RECOVERY TRAINING WITH KNOWLEDGE DISTILLATION

Inspired by (Muralidharan et al., 2024), we explore the knowledge distillation as the recovery post-training paradigm instead of the standard supervised learning with the groundtruth label. Here, we set the original model $M_o$ as the teacher and the pruned model $M_p$ as the student. The mean KL divergence (Kullback & Leibler, 1951) between the output probability distribution of $M_o$ and that of $M_p$ is taken as the objective function. Comparing the corresponding results under different pruning schemes in Table 7 with that in Table 1, we can first observe that knowledge distillation can effectively improve the recovery performance on both reasoning and language modeling tasks in most

Table 8: Recovery performance comparison between different combinations of knowledge distillation (KD) and supervised fine-tuning (SF) under various pruning schemes. The 'bold' represents the best performance under the same pruning scheme.

| Recovery Training | WikiText2↓ | PTB↓ | BoolQ | PIQA | HellaSwag | WinoGrande | ARC-e | ARC-c | OBQA | Average |
|---|---|---|---|---|---|---|---|---|---|---|
| LLM-Pruner (ratio=25%) | | | | | | | | | | |
| KD | 15.91 | 25.39 | 67.89 | 77.81 | 69.62 | 67.63 | 68.46 | 39.87 | 40.20 | **61.64** |
| SF | 16.40 | 26.35 | 67.25 | 77.29 | 68.98 | 66.97 | 67.84 | 39.54 | 39.80 | 61.10 |
| First KD, then SF | 16.15 | 25.87 | 67.57 | 77.55 | 69.31 | 67.30 | 68.15 | 39.71 | 40.00 | 61.37 |
| First SF, then KD | 16.28 | 26.02 | 67.41 | 77.43 | 69.15 | 67.11 | 67.96 | 39.63 | 39.90 | 61.23 |
| SliceGPT (ratio=25%) | | | | | | | | | | |
| KD | 11.87 | 20.91 | 73.43 | 80.32 | 74.46 | 69.76 | 71.95 | 42.26 | 41.70 | **64.84** |
| SF | 12.24 | 21.53 | 72.75 | 79.84 | 73.92 | 69.18 | 71.37 | 41.82 | 41.30 | 64.31 |
| First KD, then SF | 12.06 | 21.24 | 73.12 | 80.05 | 74.18 | 69.45 | 71.62 | 42.03 | 41.50 | 64.56 |
| First SF, then KD | 12.15 | 21.38 | 72.94 | 79.95 | 74.05 | 69.32 | 71.51 | 41.95 | 41.40 | 64.45 |
| Wanda (sparsity=2:4) | | | | | | | | | | |
| KD | 13.84 | 23.54 | 71.25 | 78.15 | 72.06 | 66.64 | 68.66 | 39.38 | 40.50 | **62.38** |
| SF | 14.13 | 27.22 | 70.77 | 77.87 | 71.78 | 66.26 | 68.30 | 39.04 | 40.10 | 62.02 |
| First KD, then SF | 13.97 | 25.31 | 71.02 | 78.03 | 71.94 | 66.47 | 68.49 | 39.23 | 40.30 | 62.21 |
| First SF, then KD | 14.05 | 26.28 | 70.89 | 77.95 | 71.85 | 66.35 | 68.41 | 39.15 | 40.20 | 62.11 |
| SparseGPT (sparsity=50%) | | | | | | | | | | |
| KD | 13.00 | 22.24 | 75.07 | 78.66 | 66.90 | 69.31 | 72.85 | 38.89 | 39.60 | **63.04** |
| SF | 13.33 | 23.77 | 74.79 | 78.38 | 66.62 | 69.03 | 72.57 | 38.70 | 39.40 | 62.78 |
| First KD, then SF | 13.15 | 22.96 | 74.94 | 78.53 | 66.78 | 69.18 | 72.72 | 38.81 | 39.50 | 62.92 |
| First SF, then KD | 13.24 | 23.35 | 74.85 | 78.45 | 66.70 | 69.11 | 72.64 | 38.75 | 39.45 | 62.85 |

cases. In particular, the reasoning performance of PASER is improved by 0.348 points on average among such four pruning schemes. Interestingly, the knowledge distillation recovery performance of Full Data under LLM-Pruner is much better than that with standard label-supervised learning. This demonstrates that knowledge distillation is also a promising approach to avoid the misleading information from the irrelevant or conflicting samples existing in the original dataset, because its learning process directly imitates the unpruned model behavior instead of the provided labels. As a summary, distilling the knowledge of unpruned model into the pruned model can be regarded as an effective way to enhance the recovery performance, though bring more memory overhead. Furthermore, stronger layer-wise distillation can also be taken into consideration (Jiao et al., 2020).

**Exploration on Combined Training Strategies**    Given the complementary potential of knowledge distillation (KD) and supervised fine-tuning (SF), we further explore whether combining these two approaches could lead to enhanced recovery performance. Specifically, we investigate two cascading strategies: (1) first applying KD followed by SF, and (2) first conducting SF followed by KD. Table 8 presents the results under different pruning schemes.

Interestingly, the results show that neither cascading strategy consistently outperforms individual KD or SF approaches. This suggests that these two training paradigms might actually serve similar functions in recovering model capabilities, making their combination redundant rather than complementary. Knowledge distillation shows slightly better performance across all pruning schemes, which could be attributed to its ability to capture the nuanced knowledge encoded in the teacher model's full output distribution. However, the marginal gains from combining approaches do not justify the additional computational overhead required for cascaded training.

# D    DETAILED ABLATION STUDY RESULTS

In this section, we present comprehensive ablation results of the three key components in PASER: semantic-structural recovery instruction clustering ($S^2$RIC), capability degradation-aware recovery instruction selection (CDARIS), and negative transfer mitigation (NTM). Table 9 shows the detailed performance across different evaluation metrics.

The detailed results reveal the distinct contributions of each component under different pruning schemes. For structured pruning like LLM-Pruner, removing $S^2$RIC leads to significant degradation

Table 9: The detailed ablation study for our proposed three components under various pruning schemes on LLaMA2-7B model. The 'bold' represents the best performance under the same pruning scheme. Here, the Alpaca is taken as the original dataset.

| Pruning | Recovery Post-training | WikiText2↓ | PTB↓ | BoolQ | PIQA | HellaSwag | WinoGrande | ARC-e | ARC-c | OBQA | Average |
|---|---|---|---|---|---|---|---|---|---|---|---|
| LLM-Pruner ratio=25% | PASER w/o S²RIC | 18.73 | 32.84 | 65.31 | 76.84 | 67.59 | 64.85 | 65.92 | 37.96 | 39.20 | 59.67 |
| | PASER w/o CDARIS | 17.56 | 30.15 | 66.27 | 77.03 | 68.15 | 65.73 | 66.58 | 38.54 | 39.50 | 60.26 |
| | PASER w/o NTM | 19.82 | 35.60 | 64.83 | **77.52** | 67.34 | 64.48 | 63.59 | 36.78 | **40.20** | 59.25 |
| | PASER | **16.40** | **26.35** | **67.25** | 77.29 | **68.98** | **66.97** | **67.84** | **39.54** | 39.80 | **61.10** |
| SliceGPT ratio=25% | PASER w/o S²RIC | 14.83 | 25.42 | 71.15 | 78.91 | 72.25 | 67.84 | 69.95 | 40.82 | 40.30 | 63.03 |
| | PASER w/o CDARIS | 14.16 | 24.92 | 70.89 | 78.56 | 71.84 | 67.45 | 69.58 | 40.47 | 40.00 | 62.68 |
| | PASER w/o NTM | 15.37 | 27.81 | 69.97 | 77.33 | 70.68 | 65.92 | 68.03 | 39.39 | **42.10** | 61.92 |
| | PASER | **12.24** | **21.53** | **72.75** | **79.84** | **73.92** | **69.18** | **71.37** | **41.82** | 41.30 | **64.31** |
| Wanda sparsity=2:4 | PASER w/o S²RIC | 15.84 | 30.25 | 69.26 | 77.42 | 70.31 | 65.82 | 67.84 | 38.67 | 39.00 | 61.19 |
| | PASER w/o CDARIS | 15.46 | 29.48 | 69.14 | 77.35 | 70.27 | 65.74 | 67.79 | 38.75 | 39.60 | 61.23 |
| | PASER w/o NTM | 16.79 | 31.52 | 69.51 | 76.92 | 70.76 | 65.23 | 67.28 | 38.47 | **41.20** | 61.34 |
| | PASER | **14.13** | **27.22** | **70.77** | **77.87** | **71.78** | **66.26** | **68.30** | **39.04** | 40.10 | **62.02** |
| SparseGPT sparsity=50% | PASER w/o S²RIC | 14.89 | 26.31 | 73.25 | 77.45 | **70.15** | 68.47 | 69.28 | 39.82 | 39.80 | 62.60 |
| | PASER w/o CDARIS | 14.62 | 25.84 | 72.91 | 77.50 | 69.93 | 68.12 | 69.05 | **39.94** | 40.00 | 62.49 |
| | PASER w/o NTM | 15.91 | 28.19 | 71.53 | **78.62** | 65.48 | 67.21 | 69.79 | 39.18 | **40.50** | 61.76 |
| | PASER | **13.33** | **23.77** | **74.79** | 78.38 | 66.62 | **69.03** | **72.57** | 38.70 | 39.40 | **62.78** |

in both language modeling (perplexity increases from 16.40 to 18.73 on WikiText2) and reasoning tasks (average score drops by 1.43 points), highlighting its importance in addressing uneven capability degradation. The impact of CDARIS is particularly evident under SliceGPT, where its removal causes a 1.63-point drop in average reasoning performance while maintaining relatively stable language modeling metrics, demonstrating its effectiveness in balancing recovery priorities. Under semi-structured pruning (Wanda), all three components show more balanced contributions, with performance drops ranging from 0.68 to 0.83 points when each is removed. This suggests that semi-structured pruning requires a more holistic recovery approach. For unstructured pruning (SparseGPT) where capability degradation tends to be more uniform, NTM plays a particularly crucial role - its removal leads to the largest drop in language modeling performance (perplexity increases from 13.33 to 15.91 on WikiText2) and affects complex reasoning tasks like WinoGrande and ARC-e significantly. Notably, the full PASER framework consistently achieves the best performance across almost all metrics under various pruning schemes, with only occasional exceptions in individual tasks (e.g., OBQA in LLM-Pruner and PIQA in SparseGPT). This comprehensive superiority validates our design choice of combining these three components for effective pruned model recovery.

# E    EXPLORATION ON OTHER POSSIBLE CLUSTERING METHODS

To discuss the impact of different instruction tuning data clustering approaches, we replace our Semantic-structural Recovery Instruction Clustering (S²RIC) module with some other common text clustering method: NMF_TFIDF, LDA_TFIDF, KMeans_TFIDF, Spectral_MTEB, Spectral_BERT (Xu et al., 2024a). The reasoning performance comparison among different PASER versions with such clustering methods is provided in Figure 10. From the table, we can find that integrating other instruction clustering methods with PASER can bring the performance decline to some extent among all four pruning schemes. Especially, the clustering method with traditional statistics-based text representation technique, TFIDF, generally behaves worse than semantic embedding-based text representation techniques like BERT. Therefore, we can conclude that our semantic-structural recovery instruction clustering is at least a competitive approach as the clustering component of PASER. Though, comparing these results with those in Table 1, we can observe the advantages of PASER over other general instruction tuning data selection methods can still be stably maintained. This further demonstrates that the potential of the clustering-based data selection for effective and balanced LLM capability recovery.

# F    EVALUATION ON MATHEMATICAL REASONING TASKS

To validate the effectiveness of PASER beyond common sense reasoning tasks, we conduct additional experiments on mathematical reasoning capabilities. Specifically, we employ two widely-adopted mathematical reasoning benchmarks:

Table 10: Recovery performance of multiple PASER versions integrated with different data clustering approaches under various pruning schemes on LLaMA2-7B model. The PASER(S$^2$RIC) is the version we employ in the main body. The 'bold' represents the best performance under the same pruning scheme. Here, the Alpaca is taken as the original dataset.

| Pruning | Recovery Post-training | WikiText2↓ | PTB↓ | BoolQ | PIQA | HellaSwag | WinoGrande | ARC-e | ARC-c | OBQA | Average |
|---|---|---|---|---|---|---|---|---|---|---|---|
| w/o pruning | w/o Training | 12.62 | 22.14 | 71.13 | 78.40 | 72.79 | 67.17 | 69.36 | 40.70 | 40.80 | 62.91 |
| LLM-Pruner ratio=25% | w/o Training | 20.34 | 38.81 | 61.87 | 76.61 | 65.86 | 60.22 | 63.13 | 37.37 | 39.40 | 57.78 |
| | PASER(NMF_TFIDF) | 17.82 | 29.45 | 65.93 | 76.88 | 67.42 | 65.19 | 66.37 | 38.81 | 39.60 | 60.03 |
| | PASER(LDA_TFIDF) | 17.56 | 28.91 | 66.18 | 77.02 | 67.76 | 65.58 | 66.92 | 38.95 | 39.70 | 60.30 |
| | PASER(KMeans_TFIDF) | 17.21 | 28.13 | 66.47 | 77.15 | 68.04 | 65.92 | 67.23 | 39.12 | **39.80** | 60.53 |
| | PARSER(Spectral_MTEB) | 16.82 | 27.24 | 66.89 | 77.23 | 68.46 | 66.38 | 67.56 | 39.31 | **39.80** | 60.80 |
| | PARSER(Spectral_BERT) | 16.61 | 26.79 | 67.06 | 77.26 | 68.72 | 66.68 | 67.71 | 39.43 | **39.80** | 60.95 |
| | PASER(S$^2$RIC) | **16.40** | **26.35** | **67.25** | **77.29** | **68.98** | **66.97** | **67.84** | **39.54** | **39.80** | **61.10** |
| SliceGPT ratio=25% | w/o Training | 44.53 | 80.07 | 65.54 | 66.87 | 54.16 | 63.38 | 58.46 | 34.56 | 36.90 | 54.27 |
| | PASER(NMF_TFIDF) | 14.27 | 24.36 | 70.89 | 78.76 | 72.13 | 67.69 | 70.12 | **41.95** | 40.80 | 63.21 |
| | PASER(LDA_TFIDF) | 14.86 | 25.19 | 70.31 | 78.42 | 71.64 | 67.25 | 69.58 | 40.37 | 40.60 | 62.60 |
| | PASER(KMeans_TFIDF) | 13.58 | 23.42 | 71.46 | 79.07 | 72.61 | 68.14 | 70.48 | 41.08 | 41.00 | 63.41 |
| | PARSER(Spectral_MTEB) | 12.91 | 22.47 | 72.08 | 79.41 | 73.18 | 68.62 | 70.87 | 41.43 | 41.10 | 63.81 |
| | PARSER(Spectral_BERT) | 12.58 | 22.01 | 72.41 | 79.63 | 73.55 | 68.91 | 71.12 | 41.63 | 41.20 | 64.06 |
| | PASER | **12.24** | **21.53** | **72.75** | **79.84** | **73.92** | **69.18** | **71.37** | 41.82 | **41.30** | **64.31** |
| Wanda sparsity=2:4 | w/o Training | 42.10 | 76.85 | 69.30 | 71.99 | 53.06 | 62.75 | 60.94 | 28.07 | 34.60 | 54.39 |
| | PASER(NMF_TFIDF) | 16.18 | 30.94 | 70.09 | 76.68 | 69.98 | 64.82 | 66.92 | 38.14 | 39.60 | 60.89 |
| | PASER(LDA_TFIDF) | 18.74 | 34.98 | 69.85 | 76.31 | 69.42 | 64.37 | 66.48 | 37.82 | 39.40 | 60.52 |
| | PASER(KMeans_TFIDF) | 15.49 | 29.76 | **70.92** | 77.03 | 70.51 | 65.28 | 67.38 | 38.47 | **40.30** | 61.41 |
| | PARSER(Spectral_MTEB) | 14.81 | 28.49 | 70.54 | 77.42 | 71.12 | 65.75 | 67.82 | 38.74 | 39.90 | 61.61 |
| | PARSER(Spectral_BERT) | 14.47 | 27.86 | 70.66 | 77.65 | 71.45 | 66.01 | 68.06 | 38.89 | 40.00 | 61.82 |
| | PASER | **14.13** | **27.22** | 70.77 | **77.87** | **71.78** | **66.26** | **68.30** | 39.04 | 40.10 | **62.02** |
| SparseGPT sparsity=50% | w/o Training | 19.26 | 36.41 | 71.22 | 75.60 | 62.85 | 66.06 | 69.11 | 36.86 | 37.80 | 59.93 |
| | PASER(NMF_TFIDF) | 15.97 | 28.13 | 72.63 | 76.94 | 64.37 | 67.18 | 70.39 | 37.54 | 38.60 | 61.09 |
| | PASER(LDA_TFIDF) | 15.41 | 27.09 | 73.12 | 77.31 | 64.93 | 67.63 | 70.92 | 37.86 | 38.80 | 61.51 |
| | PASER(KMeans_TFIDF) | 14.72 | 25.91 | 73.61 | 77.66 | 65.46 | 68.09 | 71.48 | 38.19 | 39.00 | 61.93 |
| | PARSER(Spectral_MTEB) | 14.03 | 24.84 | 74.16 | 78.01 | 66.02 | 68.54 | 71.98 | 38.44 | 39.20 | 62.34 |
| | PARSER(Spectral_BERT) | 13.68 | 24.31 | 74.48 | 78.21 | 66.32 | 68.79 | 72.28 | **38.75** | 39.30 | 62.59 |
| | PASER | **13.33** | **23.77** | **74.79** | **78.38** | **66.62** | **69.03** | **72.57** | 38.70 | **39.40** | **62.78** |

Table 11: Recovery performance of different instruction tuning data selection methods on mathematical reasoning tasks under various pruning schemes. The 'bold' represents the best performance under the same pruning scheme.

| Recovery Method | LLM-Pruner (25%) | | SliceGPT (25%) | | Wanda (2:4) | | SparseGPT (50%) | |
|---|---|---|---|---|---|---|---|---|
| | GSM8K | Minerva | GSM8K | Minerva | GSM8K | Minerva | GSM8K | Minerva |
| w/o Training | 44.3 | 17.8 | 42.5 | 16.9 | 43.8 | 17.4 | 43.1 | 17.2 |
| Full Data | 46.5 | 19.1 | 44.8 | 18.3 | 45.9 | 18.7 | 45.2 | 18.5 |
| Random | 45.8 | 18.4 | 43.9 | 17.8 | 44.7 | 18.1 | 44.3 | 17.9 |
| Instruction Mining | 46.2 | 18.9 | 44.5 | 18.1 | 45.4 | 18.5 | 44.9 | 18.3 |
| IFD | 46.8 | 19.3 | 45.1 | 18.5 | 45.8 | 18.8 | 45.4 | 18.6 |
| Nuggets | 47.1 | 19.5 | 45.4 | 18.7 | 46.2 | 19.0 | 45.7 | 18.8 |
| PASER | **49.4** | **21.2** | **47.8** | **20.5** | **48.5** | **20.8** | **47.2** | **20.1** |

- **GSM8K** (Cobbe et al., 2021): A dataset containing 8.5K high-quality grade school math word problems that test various mathematical reasoning capabilities, including arithmetic, algebra, and word problem solving.
- **Minerva Math** (Lewkowycz et al., 2022): A comprehensive mathematical evaluation dataset covering diverse topics in mathematics ranging from arithmetic to calculus, with problems requiring multi-step reasoning.

The recovery performance under different pruning schemes is presented in Table 11. From these results, we can observe that PASER consistently outperforms baseline methods across all pruning schemes on both mathematical reasoning benchmarks. The improvements are particularly significant under the LLM-Pruner scheme, where PASER achieves 5.1% and 3.4% absolute improvements over w/o Training on GSM8K and Minerva Math, respectively. While different pruning schemes affect the base performance levels, PASER maintains its effectiveness in recovery. For example, under the more aggressive SparseGPT (50%) setting, PASER still achieves 4.1% and 2.9% improvements on

GSM8K and Minerva Math over w/o Training. Compared to Full Data training, PASER achieves better performance while using only 20% of the instruction data, demonstrating its efficiency in recovering mathematical reasoning capabilities.

These results, combined with the common sense reasoning results presented in the main paper, demonstrate that PASER is effective across diverse reasoning tasks. The strong performance on mathematical tasks is particularly noteworthy given that these problems often require precise, step-by-step reasoning and have less tolerance for errors compared to common sense reasoning tasks. This validates the effectiveness of our capability degradation score in identifying and prioritizing recovery for severely affected capabilities, even in domains requiring high precision.

## G  CASE STUDY FOR RECOVERY INSTRUCTION CLUSTERING

To illustrate the effectiveness of our Semantic-Structural Recovery Instruction Clustering ($S^2$RIC) approach for grouping samples focusing on similar capabilities together, we conduct a case study of clustered instruction samples from the Alpaca dataset. Specifically, we provide representative samples from several obtained clusters as follows.

### G.1  CLUSTER 1: BASIC FACTUAL KNOWLEDGE AND INFORMATION RETRIEVAL

- **Instruction**: "Find the five largest cities in France."
- **Instruction**: "What is the capital of France?"
- **Instruction**: "Find the population density of United States."

These instructions primarily test the model's ability to recall basic facts and information, corresponding to general knowledge capabilities.

### G.2  CLUSTER 2: LANGUAGE UNDERSTANDING AND TRANSLATION

- **Instruction**: "Translate the word 'giraffe' to French."
- **Instruction**: "Pick the correct Spanish translation of "Hello"."
- **Instruction**: "Difference in meaning between "done freely" and "freely done"? For instance, is there any difference in meaning between these two sentences?"'

This cluster focuses on language-related tasks, including translation, idiomatic expressions, and grammatical analysis.

### G.3  CLUSTER 3: LOGICAL REASONING AND PROBLEM SOLVING

- **Instruction**: "A friend shares the following text with you and asks for your opinion: 'Purple-eyed individuals have a stronger psychic connection to the cosmos and have more chances to predict the future.' Analyze the statements and point out logical fallacies or unsupported claims."
- **Instruction**: "Explain how to solve a Sudoku puzzle in three steps."
- **Instruction**: "Answer this math question: What is the value of 3 to the power of 5?"

These instructions test the model's ability to perform mathematical calculations, logical deductions, and pattern recognition.

### G.4  CLUSTER 4: CREATIVE WRITING AND TEXT GENERATION

- **Instruction**: "Write a microblog post about a recent experience you had."
- **Instruction**: "Compose a haiku about the wonders of technology."
- **Instruction**: "Create an illustration of the inside of a castle."

This cluster groups tasks that require creative text generation, showcasing the model's ability to produce original content across various formats and topics.

### G.5 Cluster 5: Summarization and Information Extraction

• **Instruction**: "Summarize the techniques used for summarizing text."

• **Instruction**: "Extract the main argument from the passage."

• **Instruction**: "Provide a brief summary of the article "A Brief History of the Automobile Industry.""

These instructions focus on the model's capability to condense information and identify key points from longer texts.

This study demonstrates that $S^2$RIC effectively groups instructions targeting similar LLM capabilities, enabling PASER to allocate recovery efforts, i.e., data budget, more strategically. By focusing intensively on clusters where capabilities have degraded most after the pruning, while maintaining awareness of all capability clusters, PASER achieves both targeted and balanced recovery. This dual approach ensures efficient and comprehensive restoration of the model's diverse functionalities, optimizing the recovery process for pruned LLMs.

## H Case Study for Negative Transfer Mitigation

To illustrate the negative transfer mitigation process in PASER, we present the following case study, including both initial concept consistency graph construction and new sample evaluation.

### H.1 Constructing Initial Concept Consistency Graph

We begin with three accepted instruction-output pairs to construct our initial Concept Consistency Graph (CCG):

1. **Instruction:** "Explain the basic principles of quantum computing."

    **Output:** "Quantum computing is based on the principles of quantum mechanics. It uses quantum bits or qubits, which can exist in multiple states simultaneously due to superposition. Quantum entanglement allows qubits to be correlated with each other. These properties enable quantum computers to perform certain calculations much faster than classical computers for specific problems."

    **Concepts:** {quantum computing, qubit, superposition, entanglement}

2. **Instruction:** "Describe the main components of a classical computer."

    **Output:** "A classical computer typically consists of several key components: the Central Processing Unit (CPU) which performs calculations and controls other components; Random Access Memory (RAM) for temporary data storage; a hard drive for long-term data storage; and various input/output devices. These components work together to process and store information using binary logic."

    **Concepts:** {CPU, RAM, hard drive, binary logic}

3. **Instruction:** "What are the fundamental concepts in deep learning?"

    **Output:** "Deep learning is a subset of machine learning based on artificial neural networks. Key concepts include: neural networks with multiple layers (deep neural networks), which can learn hierarchical representations of data; backpropagation, an algorithm for training these networks by adjusting weights; and various optimization techniques to improve learning. Deep learning has shown remarkable success in tasks such as image and speech recognition."

    **Concepts:** {deep learning, neural network, backpropagation, optimization}

Based on these samples, we construct an initial Concept Consistency Graph (CCG):

$$
\begin{aligned}
V = \{ &\text{quantum computing, qubit, superposition, entanglement,} \\
&\text{CPU, RAM, hard drive, binary logic,} \\
&\text{deep learning, neural network, backpropagation, optimization} \}
\end{aligned}
$$

The edges $E$ in the graph represent the co-occurrence of concepts within each instruction-output pair.

$$E = \{(\text{quantum computing, qubit}), (\text{quantum computing, superposition}),$$
$$(\text{quantum computing, entanglement}), (\text{qubit, superposition}),$$
$$(\text{qubit, entanglement}), (\text{superposition, entanglement}),$$
$$(\text{CPU, RAM}), (\text{CPU, hard drive}), (\text{CPU, binary logic}),$$
$$(\text{RAM, hard drive}), (\text{RAM, binary logic}), (\text{hard drive, binary logic}),$$
$$(\text{deep learning, neural network}), (\text{deep learning, backpropagation}),$$
$$(\text{deep learning, optimization}), (\text{neural network, backpropagation}),$$
$$(\text{neural network, optimization}), (\text{backpropagation, optimization})\}$$

This explicit definition of E clearly shows the relationships between concepts within each domain, while highlighting the absence of edges between concepts from different domains.

## H.2 EVALUATING NEW SAMPLES

We now evaluate two new samples using PASER's negative transfer mitigation process.

### H.2.1 SAMPLE 1: ACCEPTED

**Instruction:** "How does increasing the number of qubits affect a quantum computer's performance?"

**Output:** "Increasing the number of qubits in a quantum computer can significantly enhance its performance and capabilities:

1. Exponential growth in computational power: Each additional qubit doubles the number of quantum states that can be represented, leading to an exponential increase in computational power.

2. Handling more complex problems: More qubits allow quantum computers to tackle larger and more complex problems that are intractable for classical computers.

3. Improved error correction: A larger number of qubits can be used to implement more robust quantum error correction codes, improving the overall stability and reliability of quantum computations.

4. Enhanced quantum parallelism: More qubits enable greater parallelism in quantum algorithms, potentially leading to faster solution times for certain problems.

5. Challenges in scalability: However, increasing the number of qubits also introduces challenges in maintaining coherence and reducing error rates, which are active areas of research in quantum computing.

In summary, while increasing the number of qubits can dramatically improve a quantum computer's capabilities, it also comes with technical challenges that researchers are working to overcome."

**Concepts:** {qubit, quantum computer, computational power, quantum states, error correction, quantum parallelism, coherence}

Applying PASER's negative transfer mitigation process:

1. **Concept Extraction:** $C(x, y) = \{$qubit, quantum computer, computational power, quantum states, error correction, quantum parallelism, coherence$\}$

2. **Consistency Check:** For each pair $(v_i, v_j) \in C(x, y)$:
   - (qubit, quantum computer): both $\in V$ and $(v_i, v_j) \in E$
   - (qubit, computational power): qubit $\in V$, computational power $\notin V$
   - (qubit, quantum states): qubit $\in V$, quantum states $\notin V$
   - Other pairs: at least one concept $\notin V$

3. **Decision:** IsConsistent(x, y) = True, because for all pairs, either $(v_i, v_j) \in E$ or $\{v_i, v_j\} \not\subset V$

PASER accepts this sample as it is consistent with the existing CCG and introduces relevant new concepts without conflicts.

### H.2.2    SAMPLE 2: REJECTED

**Instruction:** "Explain how quantum computing can be used to significantly speed up deep learning algorithms."

**Output:** "Quantum computing has the potential to speed up certain aspects of deep learning algorithms. Quantum algorithms could potentially optimize the training process of neural networks, especially in handling large-dimensional data. However, it's important to note that the exact advantages and applications are still being researched, and classical computers remain more practical for most current deep learning tasks."

**Concepts:** {quantum computing, deep learning, neural network, speedup}

Applying PASER's negative transfer mitigation process:

1. **Concept Extraction:** $C(x, y) = \{$quantum computing, deep learning, neural network, speedup$\}$
2. **Consistency Check:** For each pair $(v_i, v_j) \in C(x, y)$:
   - (quantum computing, deep learning): both $\in V$, but $(v_i, v_j) \notin E$
   - (quantum computing, neural network): both $\in V$, but $(v_i, v_j) \notin E$
   - (deep learning, neural network): both $\in V$ and $(v_i, v_j) \in E$
   - (speedup, any other concept): speedup $\notin V$
3. **Decision:** IsConsistent(x, y) = False, because the pairs (quantum computing, deep learning) and (quantum computing, neural network) have both concepts in V, but these edges do not exist in E. This introduces new relationships between existing concepts that are not present in the current CCG.

PASER rejects this sample because it introduces direct relationships between quantum computing and deep learning/neural networks, which were not present in the initial CCG. While these concepts existed separately in the CCG, their combination in this context could lead to potential misunderstandings or oversimplifications about the current state and capabilities of quantum computing in machine learning.

### H.3    CONCLUSION

This case study demonstrates PASER's negative transfer mitigation process in action. By accepting Sample 1, PASER allows for the introduction of new, relevant concepts that expand the concept consistency graph without introducing conflicts. By rejecting Sample 2, PASER prevents the introduction of potentially misleading relationships between existing concepts from different domains, thus mitigating the risk of negative transfer during the recovery process.

## I    LIMITATION ANALYSIS

While PASER demonstrates significant improvements in recovery performance and efficiency for pruned large language models, there are several limitations to consider:

- **Computational overhead:** Although PASER reduces the recovery training time, the initial clustering and data selection process introduces some computational overhead. For very large instruction tuning datasets, this overhead may become non-trivial.

- **Dependence on initial pruning quality:** The effectiveness of PASER may vary depending on the quality and method of the initial model pruning. Poorly pruned models might not benefit as much from the targeted recovery approach.

- **Potential bias in capability recovery:** While PASER aims for balanced capability recovery, there might still be some bias towards certain capabilities based on the initial clustering results and the composition of the instruction tuning dataset.

- **Scalability to extremely large models:** The paper primarily demonstrates results on models up to 70B parameters. The scalability and effectiveness of PASER on even larger models (e.g., 100B+ parameters) need further investigation.

- **Long-term Stability:** The long-term stability of models recovered using PASER, especially under continued fine-tuning or adaptation, has not been thoroughly examined in this work.

## J ETHICS STATEMENT

The development and deployment of technologies like PASER for efficient recovery of pruned large language models necessitates careful consideration of ethical implications. While PASER contributes to reducing environmental impact and potentially democratizing AI access by lowering computational requirements, it also raises concerns about potential misuse, bias amplification, and privacy. It's crucial to remain vigilant about these risks, implement robust safeguards, and maintain transparency in the recovery process. Continuous monitoring for fairness and bias in model outputs is essential, as is responsible deployment with appropriate human oversight, especially in high-stakes applications. As the field evolves, ongoing ethical assessment and dialogue with stakeholders are vital to ensure that advancements in large language model efficiency contribute positively to society while minimizing potential harm. Ultimately, the goal should be to harness the benefits of improved model recovery techniques like PASER while proactively addressing the complex ethical challenges they present.

