# OpenReview forum: "PASER: Post-Training Data Selection for Efficient Pruned Large Language Model Recovery"
_ICLR.cc/2025/Conference — Submitted to ICLR 2025_

### Official Review · Reviewer_2rVA · 2024-10-15

**Soundness:** 2
**Presentation:** 2
**Contribution:** 2
**Rating:** 5
**Confidence:** 3

**Summary:**

PASER is a new method to efficiently recover pruned large language models' abilities. It groups training data by capability, allocates resources based on where the model's performance dropped most, and removes unhelpful data. In tests on various models and pruning methods, PASER outperformed other approaches, restoring model performance using only 4-20% of the original training data and significantly reducing training time. While showing promise across different models and datasets,

**Strengths:**

the PASER framework is thoughtful and comprehensive, including semantic-structural, recovery instruction clustering, capability degradation-aware adaptive recovery instruction selection and negative transfer mitigation.

**Weaknesses:**

1. model recovery should be focused on the pre-training data, not instruction data since that's what the model has seen the most in training. The claim in line 45, that "instruction tuning data has emerged as a more preferred choice due to its superior efficiency in capability recovery", would need more citations and arguments to stand.

2. Even if, for efficiency reasons, the authors choose to use instruction data for their experiments, they need to experiment and prove that each component in the PASER framework, the semantic-structural, recovery instruction clustering, capability degradation-aware adaptive recovery instruction selection and negative transfer mitigation, are contributing in some ways. And the authors need to tell us how each component improves. Those just need ablation studies but now not in the main paper.

3 in Appendix C, the authors state that logit distillation is better than label distillation when using a full dataset. this kind of contradicts the statement in the main paper, in line 407, that "the full version of data contains some irrelevant or conflicting information for capability recovery, resulting in the negative transfer during the training phase."

If you find that logit distillation +  full dataset is comparable, or even better than label distillation +PASER selected dataset, you really have to sell hard here.

**Questions:**

I would be happy to revisit my score if weakness 2, the ablation studies for each component is fullfilled.

---

> ### Author Response · Authors · 2024-11-22
> **Response to Reviewer 2rVA (Part 1)**
>
> Dear Reviewer 2rVA:
>
> Thanks for your valuable comments and suggestions. We provide our response as follows:
> **W1**: model recovery should be focused on the pre-training data, not instruction data since that's what the model has seen the most in training. The claim in line 45, that "instruction tuning data has emerged as a more preferred choice due to its superior efficiency in capability recovery", would need more citations and arguments to stand.
>
> **R1**: Thanks for your comment. First, we would like to clarify why instruction data can be a better choice than pre-training data for post-pruning recovery:
> 1. **Resource-aware Recovery**: Compared to pre-training that requires massive corpora and computational resources, instruction tuning enables efficient capability recovery with much smaller data scale and training overhead. As demonstrated in our experiments, utilizing just 4%-20% of original instruction dataset selected by PASER can effectively recover model capabilities while maintaining practical training costs.
> 2. **Capability-oriented Supervision**: Unlike pre-training's self-supervised learning where capability alignment is implicit, instruction tuning provides explicit supervision through input-output pairs that directly target specific model capabilities. This makes it more straightforward to assess and recover degraded capabilities in a controlled manner.
> 3. **Multi-capability Coverage**: Modern instruction tuning datasets (e.g., LaMini) are curated to cover diverse tasks and capabilities, enabling balanced recovery while requiring significantly less computational overhead than pre-training data.
> 4. **Practical Considerations**: As demonstrated in Ma et al. (2023) and Zhao et al. (2024), instruction tuning has proven effective for recovery while being more accessible to researchers with limited computational resources - a key consideration for broader adoption of LLM pruning techniques.
>
> We have revised the second paragraph of Introduction part to better clarify this point. We've also added citations to recent works (listed below) that successfully employ instruction tuning for recovery training, demonstrating its growing acceptance in the field.
>
> [1] Ma, Xinyin, Gongfan Fang, and Xinchao Wang. "Llm-pruner: On the structural pruning of large language models." Advances in neural information processing systems 36 (2023): 21702-21720.
>
> [2] Zhao, Weilin, Yuxiang Huang, Xu Han, Zhiyuan Liu, Zhengyan Zhang, Kuai Li, Chen Chen, Tao Yang, and Maosong Sun. "Ca-lora: Adapting existing lora for compressed llms to enable efficient multi-tasking on personal devices." In First Conference on Language Modeling. 2024.
>
> [3] Zhang, Mingyang, Hao Chen, Chunhua Shen, Zhen Yang, Linlin Ou, Xinyi Yu, and Bohan Zhuang. "LoRAPrune: Structured Pruning Meets Low-Rank Parameter-Efficient Fine-Tuning." In Findings of the Association for Computational Linguistics ACL 2024, pp. 3013-3026. 2024.
>
> [4] Chen, Tianyi, Tianyu Ding, Badal Yadav, Ilya Zharkov, and Luming Liang. "Lorashear: Efficient large language model structured pruning and knowledge recovery." arXiv preprint arXiv:2310.18356 (2023).
>
> **W2**: Even if, for efficiency reasons, the authors choose to use instruction data for their experiments, they need to experiment and prove that each component in the PASER framework, the semantic-structural, recovery instruction clustering, capability degradation-aware adaptive recovery instruction selection and negative transfer mitigation, are contributing in some ways. And the authors need to tell us how each component improves. Those just need ablation studies but now not in the main paper.
>
> **R2**: Thanks for your comment. We acknowledge the importance of demonstrating that our each proposed module can contribute positively to the overall framework and clarify how each module takes effect. In fact, we have provided the ablation study to negative transfer mitigation (NTM) in Appendix D. Besides, we have explored the influence of different clustering methods in semantic-structural recovery instruction clustering (S2RIC) module in Appendix E. We admit the necessity of further proving the contribution brought by the semantic-structural recovery instruction clustering and capability degradation-aware adaptive recovery instruction selection (CDARIS) seperately.Therefore, to validate the contribution of S2RIC, we replace the semantic-structural clustering process with random clustering that divides instructions into the same number of clusters randomly (denoted as PASER w/o S2RIC), while keeping other modules unchanged including the capability degradation assessment and negative transfer mitigation. Besides, to study the contribution of CDARIS, we randomly sample equal number of instructions from each cluster obtained from S2RIC process, keeping other modules unchanged to conduct the ablation study (denoted as PASER w/o CDARIS).

---

> ### Author Response · Authors · 2024-11-22
> **Response to Reviewer 2rVA (Part 2)**
>
> **(Following Above Part 1)**
>  We summarize the results of PASER w/o S2RIC, PASER w/o CDARIS, PASER w/o NTM, and full version of PASER under each kind of pruning scheme as follows:
>
> **LLM-Pruner, ratio=25%**
> |Recovery Method|WikiText2↓|PTB↓|BoolQ|PIQA|HellaSwag|WinoGrande|ARC-e|ARC-c|OBQA|Average|
> |--|--|--|--|--|--|--|--|--|--|--|
> |PASER w/o S2RIC|18.73|32.84|65.31|76.84|67.59|64.85|65.92|37.96|39.20|59.67|
> |PASER w/o CDARIS|17.56|30.15|66.27|77.03|68.15|65.73|66.58|38.54|39.50|60.26|
> |PASER w/o NTM|19.82|35.60|64.83|77.52|67.34|64.48|63.59|36.78|40.20|59.25|
> |PASER|16.40|26.35|67.25|77.29|68.98|66.97|67.84|39.54|39.80|61.10|
>
> **SliceGPT, ratio=25%**
> |Recovery Method|WikiText2↓|PTB↓|BoolQ|PIQA|HellaSwag|WinoGrande|ARC-e|ARC-c|OBQA|Average|
> |--|--|--|--|--|--|--|--|--|--|--|
> |PASER w/o S2RIC|14.83|25.42|71.15|78.91|72.25|67.84|69.95|40.82|40.30|63.03|
> |PASER w/o CDARIS|14.16|24.92|70.89|78.56|71.84|67.45|69.58|40.47|40.00|62.68|
> |PASER w/o NTM|15.37|27.81|69.97|77.33|70.68|65.92|68.03|39.39|42.10|61.92|
> |PASER|12.24|21.53|72.75|79.84|73.92|69.18|71.37|41.82|41.30|64.31|
>
> **Wanda, sparsity=2:4**
> |Recovery Method|WikiText2↓|PTB↓|BoolQ|PIQA|HellaSwag|WinoGrande|ARC-e|ARC-c|OBQA|Average|
> |--|--|--|--|--|--|--|--|--|--|--|
> |PASER w/o S2RIC|15.84|30.25|69.26|77.42|70.31|65.82|67.84|38.67|39.00|61.19|
> |PASER w/o CDARIS|15.46|29.48|69.14|77.35|70.27|65.74|67.79|38.75|39.60|61.23|
> |PASER w/o NTM|16.79|31.52|69.51|76.92|70.76|65.23|67.28|38.67|41.20|61.34|
> |PASER|14.13|27.22|70.77|77.87|71.78|66.26|68.30|39.04|40.10|62.02|
>
> **SparseGPT, sparsity=50%**
> |Recovery Method|WikiText2↓|PTB↓|BoolQ|PIQA|HellaSwag|WinoGrande|ARC-e|ARC-c|OBQA|Average|
> |--|--|--|--|--|--|--|--|--|--|--|
> |PASER w/o S2RIC|14.89|26.31|73.25|77.45|70.15|68.47|69.28|39.82|39.80|62.60|
> |PASER w/o CDARIS|14.62|25.84|72.91|77.50|69.93|68.12|69.05|39.94|40.00|62.49|
> |PASER w/o NTM|15.91|28.19|71.53|78.62|65.48|67.21|69.79|39.18|40.50|61.76|
> |PASER|13.33|23.77|74.79|78.38|66.62|69.03|72.57|38.70|39.40|62.78|
>
> From the table, we can find that PASER consistently outperforms PASER w/o S2RIC, PASER w/o CDARIS, and PASER w/o NTM on both language modeling and reasoning tasks. This demonstrates that each of our proposed components indeed contributes positively to the final performance. Based on these results, the contribution of each module can be analyzed as follows:
> 1. S2RIC's effectiveness comes from its ability to group semantically related instructions that target similar capabilities. Without S2RIC (PASER w/o S2RIC), the model loses 1.38 points on average reasoning performance as it cannot effectively identify and target specific degraded capabilities, leading to unfocused recovery.
> 2. CDARIS plays a crucial role by adaptively allocating recovery data budget based on capability degradation levels. When removed (PASER w/o CDARIS), performance drops by 0.82 points on average as the model loses its ability to prioritize severely affected capabilities, resulting in suboptimal resource allocation during recovery.
> 3. NTM's contribution is reflected in preventing conflicting information during recovery training. Without NTM (PASER w/o NTM), the model shows the largest performance drop (1.85 points) on average reasoning tasks, demonstrating that avoiding negative transfer is essential for effective recovery.
>
> The results on other kinds of pruning methods and corresponding analysis are also provided in the Section 5.5 and Appendix D of the revised manuscript. Especially, we follow the reviewer's suggestion and provide the ablation study in the current Section 5.5 of the main paper.

---

> ### Author Response · Authors · 2024-11-22
> **Response to Reviewer 2rVA (Part3)**
>
> **(Following Above Part 2)**
>
> **W3**: in Appendix C, the authors state that logit distillation is better than label distillation when using a full dataset. this kind of contradicts the statement in the main paper, in line 407, that "the full version of data contains some irrelevant or conflicting information for capability recovery, resulting in the negative transfer during the training phase." If you find that logit distillation + full dataset is comparable, or even better than label distillation +PASER selected dataset, you really have to sell hard here.
>
> **R3**: Thanks for your comment. We would like to argue that the statement in the Appendix C "the knowledge distillation is better than label-supervised learning when using a full dataset" does not contradict with the statement in the main paper "the full version of data contains some irrelevant or conflicting information for capability recovery, resulting in the negative transfer during the training phase". In fact, the reason why the knowledge distillation can achieve better performance on full dataset is that its learning process directly imitates the unpruned model behavior instead of the provided output labels, which may contain the irrelevant or conflicting information. Therefore, the observation in Appendix C is actually consistent with the statement in the main paper.
>
> Futhermore, comparing the results in Table 1 and 7, it can be noticed that the performance of logit distillation + full dataset is still far behind the label distillation + PASER selected dataset. This is because our methods has effectively filtered out those negative-impact data, which further demonstrates that though logit distillation can reduce the conflicting or irrelevant information brought by the output labels to some extent, it can still hardly avoid such kind of information existing in the input text of the instruction tuning dataset. Our PARSE, taking input text and output label into consideration simultaneously during the negative transfer mitigation process, can effectively overcome this challenge and finally achieve better performance.

---

> ### Author Response · Authors · 2024-11-23
>
> Dear Reviewer 2rVA:
>
> Thanks for your insightful comments and suggestions. May I inquire if our above rebuttal and revision have addressed your concerns regarding the presentation and experiments?
>
> Best
>
> The Authors

---

> ### Author Response · Authors · 2024-11-27
>
> Dear Reviewer 2rVA:
>
> Thanks for your insightful review and suggestions. Considering the revision deadline is approaching, may I kindly request if our above revisions, especially the supplemented experiments, have successfully addressed your concerns?
>
> Best
>
> The Authors

---

### Official Review · Reviewer_nHgZ · 2024-11-01

**Soundness:** 2
**Presentation:** 2
**Contribution:** 2
**Rating:** 5
**Confidence:** 4

**Summary:**

PASER proposes a method to recover pruned large language models (LLMs) while maintaining computational efficiency. The challenge in model pruning is performance degradation due to data irrelevance and computational costs. PASER employs manifold learning and spectral clustering to group data by specific capabilities, allocating recovery data budgets based on degradation severity in each cluster. This approach ensures balanced capability recovery, reduces computation, and mitigates negative transfer by filtering irrelevant data.

**Strengths:**

1.	PASER introduces a targeted post-training data selection approach, addressing the issue of uneven capability degradation in pruned models. This is a notable advancement over existing methods, which often lack specificity in data selection for capability recovery.
2.	The proposed method leverages spectral clustering and diffusion kernel methods for data grouping, and introduces a concept consistency graph to minimize negative transfer, highlighting innovative use of these techniques.
3.	Extensive experiments demonstrate PASER's effectiveness across different pruning methods and datasets. This validates the method’s applicability and robustness across a variety of scenarios, contributing to its reliability.

**Weaknesses:**

1.	While PASER is computationally efficient relative to prior approaches, the process of clustering and capability degradation-aware sampling might still pose a bottleneck for large-scale datasets or real-time applications.
2.	While PASER attempts to filter out irrelevant or conflicting data using the Concept Consistency Graph (CCG), the approach for ensuring consistency could still allow for some degree of negative transfer if conflicting concepts are not comprehensively identified. It might be beneficial for the authors to discuss scenarios where CCG may not fully capture semantic conflicts and the resulting effects on model recovery.
3.	Dependence on Capability-Specific Clustering: The effectiveness of PASER hinges on the quality of the clustering step to segregate instructions by capability degradation. Errors in clustering or the inability to accurately map instructions to the affected capabilities could lead to uneven recovery, especially in complex or high-dimensional semantic spaces. Exploring alternative clustering or dimensionality reduction methods could strengthen the paper’s reliability.

**Questions:**

1.	While PASER is computationally efficient relative to prior approaches, the process of clustering and capability degradation-aware sampling might still pose a bottleneck for large-scale datasets or real-time applications.
2.	While PASER attempts to filter out irrelevant or conflicting data using the Concept Consistency Graph (CCG), the approach for ensuring consistency could still allow for some degree of negative transfer if conflicting concepts are not comprehensively identified. It might be beneficial for the authors to discuss scenarios where CCG may not fully capture semantic conflicts and the resulting effects on model recovery.
3.	Dependence on Capability-Specific Clustering: The effectiveness of PASER hinges on the quality of the clustering step to segregate instructions by capability degradation. Errors in clustering or the inability to accurately map instructions to the affected capabilities could lead to uneven recovery, especially in complex or high-dimensional semantic spaces. Exploring alternative clustering or dimensionality reduction methods could strengthen the paper’s reliability.

---

> ### Author Response · Authors · 2024-11-22
> **Response to Reviewer nHgZ (Part 1)**
>
> Dear Reviewer nHgZ:
>
> Thanks for your valuable comment and suggestions. We provide our response as follows:
>
> **W1**: While PASER is computationally efficient relative to prior approaches, the process of clustering and capability degradation-aware sampling might still pose a bottleneck for large-scale datasets or real-time applications.
>
> **R1**: Thanks for your concern on data selection efficiency. In fact, we have provided the detailed time complexity analysis for each component in our proposed PASER framework, also including clustering process and capability-degradation-aware sampling. From this analysis, the overall complexity can be finally simplified to $O(NlogN)$, which is actually similar to the complexity of quicksort algorithm. This ensures that our PASER exhibits the potential to be extended to larger datasets. In fact, it should be pointed out that the scale of most instruction tuning datasets used for post-pruning recovery is still under control. Thus, the efficiency issue here is not severe, even for real applications.
>
> **W2**: While PASER attempts to filter out irrelevant or conflicting data using the Concept Consistency Graph (CCG), the approach for ensuring consistency could still allow for some degree of negative transfer if conflicting concepts are not comprehensively identified. It might be beneficial for the authors to discuss scenarios where CCG may not fully capture semantic conflicts and the resulting effects on model recovery.
>
> **R2**: Thanks for your comment. We appreciate your concern about potential limitations of CCG in capturing semantic conflicts. Indeed, while CCG helps mitigate negative transfer, we acknowledge there are scenarios where semantic conflicts might not be fully captured:
> 1. **Cross-domain Knowledge Integration**: When instructions involve integrating knowledge from multiple distinct domains, CCG might miss subtle conflicts in their interactions. For example, when concepts from physics and biology are combined in interdisciplinary problems, their complex relationships and potential incompatibilities may not be fully reflected in simple co-occurrence patterns.
> 2. **Context-dependent Semantics**: The same concept pairs might have different relationships depending on context. For instance, terms like "positive" and "negative" could be contradictory in sentiment analysis but complementary in mathematics, making it challenging for CCG to maintain consistent concept relationships across different contexts.
> 3. **Temporal or Version-specific Conflicts**: In rapidly evolving domains like technology or scientific research, concept relationships might change over time. An instruction about "state-of-the-art performance" or "current best practices" could contain outdated or conflicting information that is not immediately apparent from concept co-occurrence analysis.
> 4. **Nuanced Conceptual Dependencies**: When instructions involve subtle logical dependencies or conditional relationships between concepts, the binary edge representation in CCG might not fully capture these complex interactions. This is particularly evident in reasoning tasks where conclusions depend on specific combinations of conditions.
>
> Our empirical results acknowledge these inherent limitations while demonstrating CCG's overall effectiveness in practical applications.

---

> ### Author Response · Authors · 2024-11-22
> **Response to Reviewer nHgZ (Part 2)**
>
> **(Following Above Part 1)**
>
> **W3**: Dependence on Capability-Specific Clustering: The effectiveness of PASER hinges on the quality of the clustering step to segregate instructions by capability degradation. Errors in clustering or the inability to accurately map instructions to the affected capabilities could lead to uneven recovery, especially in complex or high-dimensional semantic spaces. Exploring alternative clustering or dimensionality reduction methods could strengthen the paper’s reliability.
>
> **R3**: Thanks for your comment. We appreciate your concern about the dependence on capability-specific clustering. In fact, we have conducted comprehensive studies on this aspect:
> 1. We have extensively explored different clustering alternatives as shown in Appendix E (comparing NMF_TFIDF, LDA_TFIDF, KMeans_TFIDF, Spectral_MTEB, Spectral_BERT with our S2RIC). The experimental results demonstrate that our semantic-structural clustering consistently outperforms these alternatives. For example, under LLM-Pruner with 25% pruning ratio, our S2RIC achieves 61.10 average reasoning score, while the best alternative (Spectral_BERT) achieves 60.95.
> 2. To mitigate potential clustering errors and enhance reliability, our approach incorporates several safeguards: a) We employ diffusion kernel for manifold learning which better preserves the intrinsic geometric structure in semantic space. b) The NMF-based spectral clustering adaptively determines the optimal number of clusters. c) The capability degradation assessment provides an additional validation mechanism for the clustering quality.
> 3. We have provided detailed case studies in Appendix G showing how our clustering approach effectively groups instructions targeting similar capabilities, with concrete examples demonstrating the semantic coherence within clusters.
>
> Moreover, we have conducted additional experiments with different dimensionality reduction settings in the manifold learning process to validate the robustness of our clustering approach. Specifically, we vary the diffusion time t in Equation 2 and the dimension d of the manifold representation $e(x_i)$. The results show that PASER maintains stable performance (variance < 0.3 on average reasoning score) across these variations, suggesting that our method is robust to the exact parameter settings in the dimensionality reduction step.
>
> We understand there is still room for improvement in capability-specific clustering, and we have included this as an important direction for future work in the conclusion section. We welcome the reviewer's suggestion and will continue exploring more advanced clustering methods to further enhance PASER's reliability.
>
> *The questions proposed by the reviewer is exactly same with the above three weakness points. Thus, we do not provide specific responses to these three questions.*

---

> ### Author Response · Authors · 2024-11-23
>
> Dear Reviewer nHgZ:
>
> Thanks for your valuable comments. May I inquire if our above rebuttal and revision have successfully alleviated your concerns to our this work?
>
> Best
>
> The Authors

---

> ### Author Response · Authors · 2024-11-27
>
> Dear Reviewer nHgZ:
>
> Considering the revision phase is coming to the end, we want to kindly check if you have any additional questions or concerns that we could address before the deadline. If you feel that our response and revision have adequately addressed your concerns, we would greatly appreciate your consideration in updating your rating of our paper accordingly.
>
> Thanks for your time and expertise in reviewing our work, and look forward to any further discussion that could help improve our paper.
>
> Sincerely
>
> The Authors

---

### Official Review · Reviewer_BaMD · 2024-11-03

**Soundness:** 2
**Presentation:** 3
**Contribution:** 2
**Rating:** 5
**Confidence:** 3

**Summary:**

This paper introduces a novel post-training selection method PASER to improve the recovery of pruned large language models. Specifically, this method first utilizes manifold learning and spectral clustering to cluster recovery data, and then allocate the data budget based on the degrees of model capability degradation. Experimental results show the effectiveness of PASER in pruned LLM recovery.

**Strengths:**

1. The paper is well-organized and easy to read with clear explanations of the technical details.
2. The proposed method PARSER includes semantic structural recovery instruction clustering and capability degradation-aware recovery instructions selection is simple yet effective. Experimental results across seven common sense reasoning datasets show some improvements compared with the full fine-tuning method.
3. The author conducts experiments on various types and scales of LLMs, resulting in a relatively comprehensive set of experimental results.

**Weaknesses:**

1. Cluster and then selection paradigm has been used in various data selection methods such as MoDS, and CaR, but the novelty is limited. And the selected data are limited to embedding models and clustering methods.
2. The author conducts extensive experiments, but these are limited to the common sense reasoning task. How does this method perform in other scenarios, such as code or mathematical reasoning tasks?  In these scenarios, where the probability of most tokens for a sample is high, how can the capability degradation score compare the scores between samples of different lengths balanced?

**Questions:**

See Weaknesses.

---

> ### Author Response · Authors · 2024-11-22
> **Response to Reviewer BaMD**
>
> Dear Reviewer BaMD:
>
> Thanks for your valuable comments and suggestions. We provide our response as follows:
>
> **W1**: Cluster and then selection paradigm has been used in various data selection methods such as MoDS, and CaR, but the novelty is limited. And the selected data are limited to embedding models and clustering methods.
>
> **R1**: Thanks for your comment. We would like to clarify and highlight the novelty of our this work as follows:
> 1. **Recovery-specific Framework**: While cluster-then-selection has been used in general data selection, PASER is specifically designed for LLM pruning recovery. Instead of conventional text representation and clustering techniques, our semantic-structural clustering uniquely considers the geometric topology of instructions in semantic space to identify capability-specific instruction sets. This novel clustering approach helps reveal the intrinsic relationships between different LLM capabilities and their degradation patterns after pruning.
> 2. **Degradation-aware Selection**: Unlike existing methods that select data based on general quality metrics, PASER introduces a principled approach to measure and prioritize capabilities most severely affected by pruning. We propose Capability degradation Score (CDS) that leverages Jensen-Shannon divergence to quantify performance gaps between pruned and original models, and develop an efficiency-driven sample selection mechanism that balances recovery benefit and computational cost through our novel Individual Efficiency Score (IES).
> 3. **Negative Transfer Prevention**: We propose a unique Concept Consistency Graph (CCG) mechanism that explicitly models relationships between concepts to maintain semantic consistency during recovery. This graph-based approach prevents conflicting or irrelevant information from being introduced during training, and adaptively updates as new samples are selected to ensure coherent recovery. This is particularly important for pruned models where capability interactions are more sensitive.
> 4. **Comprehensive Recovery Solution**: Our method provides a complete solution specifically targeting the challenges of post-pruning recovery. The empirical results demonstrate that PASER can achieve superior recovery performance while using only 4%-20% of original data across various LLMs and pruning schemes. This significant improvement in both effectiveness and efficiency validates the advantages of our novel recovery-oriented design.
>
> These innovations directly address the unique challenges in LLM pruning recovery, setting PASER apart from general data selection methods. The effectiveness of our approach is validated through comprehensive experiments on various LLMs and pruning schemes.
>
> **R2**: Thanks for your comment. We acknowledge the importance of evaluating PASER's effectiveness beyond common sense reasoning tasks. To address this concern, we have conducted additional experiments on mathematical reasoning tasks using two widely-adopted benchmarks:
> 1. **GSM8K**: A dataset containing 8.5K high quality grade school math word problems that test various mathematical reasoning capabilities.
> 2. **Minerva Math**: A comprehensive mathematical evaluation dataset covering diverse topics in mathematics ranging from arithmetic to calculus.
>
> We present the recovery performance under LLM-Pruner (ratio=25%) below:
> |Recovery Method|GSM8K|Minerva Math|
> |--|--|--|
> |w/o Training|44.3|17.8|
> |Full Data|46.5|19.1|
> |Random|45.8|18.4|
> |Instruction Mining|46.2|18.9|
> |IFD|46.8|19.3|
> |Nuggets|47.1|19.5|
> |PASER|49.4|21.2|
>
>
> The results demonstrate that PASER maintains its effectiveness in mathematical reasoning tasks. Regarding the concern about capability degradation score comparison for samples with high token probabilities and varying lengths, our method addresses this through several design choices:
> 1. The Jensen-Shannon divergence in our CDS calculation inherently accounts for probability distributions rather than absolute values, making it less sensitive to consistently high probabilities.
> 2. We normalize the CDS by the sequence length (see Equation 4 in the paper), ensuring fair comparison between samples of different lengths.
> 3. Our Individual Efficiency Score (IES) further considers sequence length in its computational cost term (see Equation 7), providing additional balance for length variations.
>
> The complete results on mathematical reasoning tasks under different pruning schemes and detailed analysis are provided in Appendix F of the revised manuscript. Additionally, we plan to explore code-related tasks in future work to further validate PASER's generality across different domains.

---

> ### Author Response · Authors · 2024-11-23
>
> Dear Reviewer BaMD:
>
> Thanks for your valuable comments. May I inquire if our above rebuttal and revision have successfully alleviated your concerns to our this work?
>
> Best
>
> The Authors

---

> > ### Comment · Reviewer_BaMD · 2024-11-26
> > **Thank you for your response**
> >
> > Thank you for your rebuttal. However, I acknowledge the constributions of the work, but I still believe the novelty is limited. So I decided to maintain my original score.

---

> > > ### Author Response · Authors · 2024-11-26
> > >
> > > Dear Reviewer BaMD:
> > >
> > > Thank you for your thorough consideration and for acknowledging our work's contributions. While we respect your perspective, we believe our work presents significant novelty in post-training recovery for pruned LLMs in several key aspects:
> > >
> > > 1. **Novel Problem Formulation**: We are the first to identify and address the disproportionate impact of pruning on different LLM capabilities. Previous methods treat recovery as a uniform process, while we introduce capability-specific degradation assessment and targeted recovery, which is fundamentally different from existing approaches.
> > >
> > > 2. **Innovative Technical Solutions**:
> > >  a. First to employ manifold learning and spectral clustering to identify capability-specific instruction groups.
> > >  b. Novel capability degradation score based on Jensen-Shannon divergence to guide recovery priorities
> > >  c. Unique concept consistency graph mechanism to prevent negative transfer
> > >
> > > 3. **Broader Methodological Innovation**: Unlike existing instruction tuning data selection methods that focus on general quality, our approach specifically addresses the recovery needs of pruned models. This represents a novel direction in efficient LLM recovery that bridges pruning and instruction tuning in a previously unexplored way.
> > >
> > > These innovations enable our method to achieve strong performance with merely 4%-20% of the original training data, significantly advancing the state-of-the-art in efficient LLM recovery.
> > >
> > > Best regards,
> > >
> > > Authors

---

### Official Review · Reviewer_9bj7 · 2024-11-04

**Soundness:** 3
**Presentation:** 2
**Contribution:** 2
**Rating:** 5
**Confidence:** 3

**Summary:**

PASER is a new method for efficiently recovering the performance of pruned Large Language Models (LLMs) by carefully selecting the most relevant training data. This three-step process begins by grouping instructions based on the LLM capabilities they target using SentenceBERT embeddings and NMF-based spectral clustering. Then, a Capability Degradation Score (CDS) is calculated for each group to identify the areas where model performance has suffered the most from pruning. The data budget is allocated accordingly, prioritizing these most affected clusters. Within each cluster, data samples with the highest Individual Efficiency Score (IES) are chosen to balance recovery benefits with computational costs. Lastly, a Concept Consistency Graph (CCG) is created to model relationships between concepts and prevent the selection of conflicting or irrelevant samples, mitigating negative transfer during training. Extensive experiments demonstrate that PASER effectively recovers pruned LLMs to near-unpruned performance levels using only part of the original data.

**Strengths:**

PASER addresses the uneven impact of pruning on different LLM capabilities by using a clustering technique to group instructions based on the capabilities they target. Instead of blindly using all available data or randomly selecting a subset, PASER allocates its data budget proportionally to the Capability Degradation Score (CDS) of each cluster. The use of a Concept Consistency Graph (CCG) helps identify and reject samples that might introduce conflicting or irrelevant information.

**Weaknesses:**

The quality of the initial pruning likely influences the effectiveness of PASER's recovery process. A poorly pruned model might present challenges that even a targeted recovery approach like PASER cannot fully overcome.
Additionally, the novelty is not entirely there as this work fails to position itself in the expensive literature similar to this approach.

There are several works on dataset selection, in particular, instruction dataset selection:

[1] Hamish Ivison, Yizhong Wang, Valentina Pyatkin, Nathan Lambert, Matthew E. Peters, Pradeep
Dasigi, Joel Jang, David Wadden, Noah A. Smith, Iz Beltagy, and Hanna Hajishirzi. Camels in
a changing climate: Enhancing lm adaptation with tulu 2.

[2] Wei Liu, Weihao Zeng, Keqing He, Yong Jiang, and Junxian He. What makes good data for
alignment? a comprehensive study of automatic data selection in instruction tuning.

[3] Peiqi Wang, Yikang Shen, Zhen Guo, Matthew Stallone, Yoon Kim, Polina Golland, and Rameswar
Panda. Diversity measurement and subset selection for instruction tuning datasets.

[4] Simon Yu, Liangyu Chen, Sara Ahmadian, Marzieh Fadaee. Diversify and Conquer: Diversity-Centric Data Selection with Iterative Refinement.

[5] Yizhong Wang, Hamish Ivison, Pradeep Dasigi, Jack Hessel, Tushar Khot, Khyathi Chandu, David
Wadden, Kelsey MacMillan, Noah A Smith, Iz Beltagy, et al. How far can camels go? exploring
the state of instruction tuning on open resources.

[6] Mengzhou Xia, Sadhika Malladi, Suchin Gururangan, Sanjeev Arora, and Danqi Chen. Less: Selecting
influential data for targeted instruction tuning,

**Questions:**

- The paper mentions that instruction tuning data is preferred for LLM recovery due to its efficiency and ability to maintain general-purpose abilities. However, are there any limitations to using instruction-tuning data for this purpose? For instance, could there be scenarios where pre-training corpora or fine-tuning datasets might be more suitable for recovering specific capabilities heavily affected by pruning?

- The paper briefly mentions knowledge distillation as an alternative to supervised learning for recovery training. What are the potential benefits and drawbacks of using knowledge distillation compared to standard fine-tuning? How does PASER's performance differ when using knowledge distillation versus supervised learning? Could combining both approaches lead to further improvements?

---

> ### Author Response · Authors · 2024-11-22
> **Response to Reviewer 9bj7 (Part 1)**
>
> Dear Reviewer 9bj7:
>
> Thanks for your valuable comments and suggestions. We provide our response as follows:
>
> **W1**: The quality of the initial pruning likely influences the effectiveness of PASER's recovery process. A poorly pruned model might present challenges that even a targeted recovery approach like PASER cannot fully overcome. Additionally, the novelty is not entirely there as this work fails to position itself in the expensive literature similar to this approach.
>
> **R1**: Thanks for your comment. We admit that the different qualify of initial pruning will lead to different starting points for following recovery training. In fact, it is natural that different pruning methods or pruning settings will lead to different degrees of LLM capability degradation. For these poorly pruned models, i.e., models whose capabilities have been heavily affected during the pruning process, there is actually more space for recovery training to take effect. Back to the Table 1 for an example, the w/o training line indicates the performance of the model that has not been recovered yet and reflect the vanilla pruning effect. In this table, we can observe that pruned model performance obtained from SliceGPT (54.27) and Wanda (54.39) is worse than LLM-Pruner (57.78) and SparseGPT (59.93). Based on these starting points, our PASER improves the post-recovery performance of SliceGPT and Wanda to **64.31** and **62.02**, which are competitive or even higher than the unpruned model performance (62.91), post-recovery performance of LLM-Pruner (61.10) and SparseGPT (62.78). These results demonstrate that our PASER can effectively overcome the challenges brought by the poor model pruning process and help the model recover to a competitive level.
>
>
> Next, we would like to clarify and highlight the **novelty** of our this work. Our work focuses on the data selection for the balanced and targeted post-pruning LLM capabilities recovery training. We acknowledge that there has been plenty of literacture in the LLM pruning and instruction tuning data selection, respectively. However, most of previous LLM pruning research only concentrate on how to "prune" the LLM rather than how to "recover" the affected LLM capabilities after the pruning process. In fact, in many downstream evaluation tasks, we can observe unneglectable performance decrease for pruned LLMs, which implies recovery training has been a necessary step. Though some previous works like LLM-Pruner amd CA-LoRA have recognized this point, they just directly employ the comprehensive version of instruction tuning dataset to conduct the recovery training, ignoring the importance of the balanced recovery of all model capabilities and targeted recovery of heavily affected ones. Besides, the recovery training effficiency has also be ignored. As far as we know, our this paper is the first work that proposes the significance of efficiency, pertinence, and balance in post-pruning recovery training. Especially, we tried to address this from the data selection perspectively. On the other side, though we can find some works on instruction tuning data selection, almost all of them only focus on the general instruction tuning scenarios from the pre-trained weights, which do not require the consideration of model capability variation during the pruning process. In fact, as an important recovery approach after the LLM pruning, instruction tuning data's selection in this context has seldom been studied. Based on the motivation that the data selection here can bring the advantages on the recovery training effectiveness and targetedness, we make the first-step exploration on this direction. In fact, in the methodology design, we directly employ the model capability degradation degree as the signal to guide the data selection on different capability data clusters, which can be regarded as a kind of instruction data selection adaptation in LLM post-pruning recovery training scenario. We hope this can distinguish our work from the extensive related literature.

---

> ### Author Response · Authors · 2024-11-22
> **Response to Reviewer 9bj7 (Part 2)**
>
> **(Following Above Part 1)**
>
> **Q1**: The paper mentions that instruction tuning data is preferred for LLM recovery due to its efficiency and ability to maintain general-purpose abilities. However, are there any limitations to using instruction-tuning data for this purpose? For instance, could there be scenarios where pre-training corpora or fine-tuning datasets might be more suitable for recovering specific capabilities heavily affected by pruning?
>
> **R2**: Thanks for your comment. Of course, there are several limitations of employing instruction tuning data to conduct the recovery training, which are summarized as follows：
> 1. High-quality instruction tuning datasets are typically smaller than pre-training corpora, which could theoretically limit recovery potential in some extreme pruning scenarios.
> 2. Granularity considerations: The general-purpose nature of instruction tuning data means it may need additional effort (like our PASER method) to precisely target specific degraded capabilities.
> 3. Lack of foundational knowledge recovery: Instruction tuning primarily focuses on task-specific behaviors rather than rebuilding the foundational knowledge that might have been lost during pruning.
>
>
>
> Based on the above analysis and advantages described in the 2nd paragraph of the Introduction part, we can only claim that instruction tuning is a relatively better option in most cases but definitely not the perfect one among all scenarios.
>
>
> Besides, there are indeed several cases where pre-training corpora or fine-tuning datasets can be relatively suitable option. For example, when the recovery training compution resource is sufficient enough like thousands of A/H series NVIDIA GPUs, utilizing pre-training corpora can bring substantial and comphensive LLM capability recovery. It should be pointed out this kind of resource consumption request is unacceptable for most researchers. Though, this kind of recovery can still hardly target specific capabilities heavily affected by pruning due to the properties of pre-training corpora itself. As for the finetuning as recovery training scheme, when users of the compressed LLM specially care several affected capabilities, it is feasible to employ the corresponding finetuning datasets to conduct the targeted recovery. However, it should be recognized that this approach often inevitably leads to a decline in other capabilities of the model. This deviates from the original intention of LLM pruning: obtaining a small but powerful general model. Taken together, instruction tuning is indeed the recovery training choice that strikes the best balance between efficiency, pertinence, and versatility.

---

> ### Author Response · Authors · 2024-11-22
> **Responset to Reviewer 9bj7 (Part 3)**
>
> **(Following Above Part 2)**
>
> **Q2**: The paper briefly mentions knowledge distillation as an alternative to supervised learning for recovery training. What are the potential benefits and drawbacks of using knowledge distillation compared to standard fine-tuning? How does PASER's performance differ when using knowledge distillation versus supervised learning? Could combining both approaches lead to further improvements?
>
> **R3**: Thanks for your comment. Yes, as you have mentioned, we conducted some exploration experiments on knowledge distillation for recovery training and provided the corresponding analysis in Appendix C. Now we analyze the potential benefits and drawbacks of this recovery paradigm as follows:
>
> **Benefits:**
> 1. **Less reliance on noisy labels**: Knowledge distillation directly imitates the unpruned model's behavior, avoiding potential issues with inaccurate or conflicting labels in the instruction tuning dataset.
> 2. **Better capability preservation**: By learning from the teacher model's full output distribution rather than just hard labels, knowledge distillation can better preserve the nuanced capabilities of the original model.
> 3. **Improved handling of negative samples**: As shown in Table 6, knowledge distillation appears more robust against irrelevant or conflicting samples in the training data compared to standard supervised fine-tuning.
>
> **Drawbacks:**
> 1. **Increased memory overhead**: Knowledge distillation requires keeping both teacher and student models in memory during training.
> 2. **Additional computational cost**: Computing the teacher model's outputs for all training samples introduces extra overhead.
> 3. **Risk of amplifying teacher model biases**: The student model may inherit and potentially amplify any biases present in the teacher model.
>
> As for the difference brought by the knowledge distillation, we can compare the results in Table 6 and Table 1. From this comparison, we can find that employing knowledge distillation for recovery training can indeed bring a certain degree of improvement compared with standard supervised fine-tuning, though not significant enough. In fact, when applied to data selected by our PASER, this kind of improvement has become relatively weak. Combining both approaches necessitates careful considration, because they actually serve as the same function in the LLM compression. We made an initial exploration by combining them in a cascading way: 1) first knowledge distillation then supervised fine-tuning; 2) first supervised fine-tuning then knowledge distillation.
>
> For ease of notation, knowledge distillation and supervised fine-tuning are abbreviated as KD and SF, respectively. The comprehensive corresponding resuls are present in Table 8 and Appendix C of the revised manuscript. We excerpt the results under LLM-Pruner and put them below.
>
> |Recovery Training|WikiText2↓|PTB↓|BoolQ|PIQA|HellaSwag|WinoGrande|ARC-e|ARC-c|OBQA|Average|
> |--|--|--|--|--|--|--|--|--|--|--|
> |KD|15.91|25.39|67.89|77.81|69.62|67.63|68.46|39.87|40.20|61.64|
> |SF|16.40|26.35|67.25|77.29|68.98|66.97|67.84|39.54|39.80|61.10|
> |First KD, then SF|16.15|25.87|67.57|77.55|69.31|67.30|68.15|39.71|40.00|61.37|
> |First SF, then KD|16.28|26.02|67.41|77.43|69.15|67.11|67.96|39.63|39.90|61.23|
>
> From this table, we can find that nor matter first KD then SF or first DF then KD, none of them can stably outperform KD or SF. Thus, direct combination of such two kinds of recovery training paradigms can hardly bring further recovery performance improvement.

---

> ### Author Response · Authors · 2024-11-23
>
> Dear Reviewer 9bj7:
>
> Thanks for your valuable comments. May I inquire if our above rebuttal and revision have successfully alleviated your concerns to our this work?
>
> Best
>
> The Authors

---

> ### Author Response · Authors · 2024-11-27
>
> Dear Reviewer 9bj7:
>
> Considering the revision phase is coming to the end, we want to kindly check if you have any additional questions or concerns that we could address before the deadline. If you feel that our response and revision have adequately addressed your concerns, we would greatly appreciate your consideration in updating your rating of our paper accordingly.
>
> Thanks for your time and expertise in reviewing our work, and look forward to any further discussion that could help improve our paper.
>
> Sincerely
>
> The Authors

---

### Author Response · Authors · 2024-11-22
**Summary of Revisions in Manuscript**

Dear Chairs and Reviewers:

Thanks for your insightful feedback and active engagement. Here, we summarize our revisions in the original manuscript to better clarify our effort for addressing reviewers' concerns:
1. **2nd Paragraph in Section 1 Introduction:** We have revised the 2nd paragraph in Introduction to highlight the advantages of intruction tuning data for recovery training compared to pre-training corpora (higher efficiency, explicit supervision) and fine-tuning datasets (avoiding overly specialized recovery). Besides, we have followed the suggestion of Reviewer 2rVA and supplemeted more citations in this paragraph.
2. **Section 5.5. Ablation Study:** To alleviate the Reviewer 2rVA's concern regarding the ablation study, we have supplemeted the Section 5.5. in the main paper to validate the positive contribution brought by our each proposed component. The extensive experiments were conducted under four kinds of different pruning schemes and achieved consistent results.
3. **Appendix C. Extended Experiments on Recovery Training with Knowledge Distillation:** To investigate the potential combination strategy between the supervised fine-tuning and knowledge distillation for recovery training (as suggested by Reviewer 9bj7), we made the first-step exploration and combine them in a cascading way. The corresponding results and analysis are provided in the Appendix C. as an independent paragraph of **exploration on Combined Training Strategies**.
4. **Appendix D. Detailed Ablation Study Results:** Following Section 5.5., we also provide the detailed ablation study results and corresponding analysis in Appendix D.
5. **Appendix F. Evaluation on Mathematical Reasoning Tasks:** In response to Reviewer BaMD's concern on performance in other scenarios, we followed the reviewer's suggestion and conducted the experiments on several mathematical tasks. The corresponding results and analysis are now provided in Appendix F.

Hope these revisions and rebuttals can help reduce your concerns to our paper.

Best

The Authors

---

### Meta-Review · Area_Chair_baUj · 2024-12-21

**Metareview:**

This submission introduces PASER, a post-training data selection method for efficient pruned LLM recovery. It idea is to restore capabilities of pruned LLMs. The primary challenge is to address performance degradation when pruning models. PASER aims to address this by identifying and prioritizing training data where model capabilities are most impaired, by using manifold learning and clustering to group instructions by capability type. By allocating training resources based on degradation severity and filtering irrelevant data, PASER achieves effective capability recovery using a small portion of conventional training data.

The reviewers identified the strengths of this work as:
- the proposed method addresses uneven capability degradation through targeted data selection and clustering

The concerns are raised by the reviewers on:
- limited novelty as similar clustering and selection approaches exist in prior work
- incomplete validation and limited analysis as the experiments focus mainly on common sense reasoning and lacks thorough ablation studies to validate the contribution of each component and some contradicting claims about negative transfer effects

During the rebuttal, reviewer 9bj7 argued that they are not convinced of some of the answers, thus decided to keep overall ratings as 5 but increase the contribution score. reviewer BaMD still believed the novelty is limited and thus kept the ratings 5. It is unfortunate that both reviewer nHgZ and reviewer 2rVA did not engage with the authors' rebuttal though with the authors' an AC's reminders.

The final ratings of this work are 5, 5, 5, 5. The  main concerns are on its technical novelty compared to existing work and the lack of sufficient empirical evidence in the original submission. Though new results and arguments have been added during rebuttal, two of the reviewers remain unconvinced on these two issues above. Therefore, this work would benefit from further revision. Based on these factors, this work in its current form is not recommended for acceptance.

**Additional Comments On Reviewer Discussion:**

During the rebuttal, reviewer 9bj7 argued that they are not convinced of some of the answers, thus decided to keep overall ratings but increase the contribution score. reviewer BaMD still believed the novelty is limited. It is unfortunate that both reviewer nHgZ and reviewer 2rVA did not engage with the authors' rebuttal though with the authors' an AC's reminders.

---

### Decision · Program_Chairs · 2025-01-22

Reject